# Sub-second ultrafast yet programmable wet-chemical synthesis

Lin Zhang[1,2], Li Peng[2], Yuanchao Lu[3], Xin Ming[2], Yuxin Sun[1], Xiaoyi Xu[2], Yuxing Xia[2], Kai Pang[2], Wenzhang Fang[2], Ning Huang[2], Zhen Xu[2,4], Yibin Ying[1], Yingjun Liu[2,4] ✉, Yingchun Fu[1] ✉ & Chao Gao[2,4] ✉

Wet-chemical synthesis via heating bulk solution is powerful to obtain nanomaterials. However, it still suffers from limited reaction rate, controllability, and massive consumption of energy/reactants, particularly for the synthesis on specific substrates. Herein, we present an innovative wet-interfacial Joule heating (WIJH) approach to synthesize various nanomaterials in a sub-second ultrafast, programmable, and energy/reactant-saving manner. In the WIJH, Joule heat generated by the graphene film (GF) is confined at the substrate-solution interface. Accompanied by instantaneous evaporation of the solvent, the temperature is steeply improved and the precursors are concentrated, thereby synergistically accelerating and controlling the nucleation and growth of nanomaterials on the substrate. WIJH leads to a record high crystallization rate of HKUST-1 (~1.97 $\mu m\,s^{-1}$), an ultralow energy cost ($9.55 \times 10^{-6}\,kWh\,cm^{-2}$) and low precursor concentrations, which are up to 5 orders of magnitude faster, −6 and −2 orders of magnitude lower than traditional methods, respectively. Moreover, WIJH could handily customize the products' amount, size, and morphology via programming the electrified procedures. The as-prepared HKUST-1/GF enables the Joule-heating-controllable and low-energy-required capture and liberation towards $CO_2$. This study opens up a new methodology towards the superefficient synthesis of nanomaterials and solvent-involved Joule heating.

Scientists have long-term pursued to use wet chemistry to synthesize nanomaterials in a faster and well-controlled manner, particularly for the synthesis on specific substrates for practical application scenarios[1–4]. Heat is an essential element in reaction thermodynamics and kinetics, and thus, thermal modulation could manipulate the synthesis towards the expected rates, pathways, and products. For instance, Joule heating technology has recently initiated a new era of synthesis[5–8]. Based on powerful heat output with high spatiotemporal resolution, it enables a series of generally unachievable processes in solid-solid and/or solid-gas systems, such as the synthesis of high-entropy nanoparticles[9], gram-scale graphene in milliseconds[10], and value-added $C_2$ products with high selectivity[11]. However, solvent-involved wet-chemical synthesis, one of the most classic methods to prepare almost all kinds of nanomaterials, has rarely benefited, mainly due to the weakened heat effect in bulk solution. Basically, wet-chemical syntheses are accomplished by heating bulk solution of precursors using autoclaves, electric or microwave ovens, et al.[12–14], in which bulk solution serves as the critical thermal medium that determines the

[1]College of Biosystems Engineering and Food Science, Key Laboratory of Intelligent Equipment and Robotics for Agriculture of Zhejiang Province, Zhejiang University, Hangzhou 310058, China. [2]MOE Key Laboratory of Macromolecular Synthesis and Functionalization, International Research Center for X Polymers, Department of Polymer Science and Engineering, Zhejiang University, Hangzhou 310027, China. [3]College of Food Science and Technology, Zhejiang University of Technology, Hangzhou 310014, China. [4]Shanxi-Zheda Institute of Advanced Materials and Chemical Engineering, Taiyuan 030032, China. ✉e-mail: yingjunliu@zju.edu.cn; ycfu@zju.edu.cn; chaogao@zju.edu.cn

reaction thermodynamics and kinetics. The bulk solution features a low boiling point and high heat capacity, making it challenging to achieve higher temperature, higher ramping/cooling rate, and thermal modulation (Fig. 1a). Hence, these traditional bulk-heating-based approaches generally suffer from long reaction time (several hours to days), poor controllability, and massive consumption of energy and reactants. These drawbacks provide an incentive to develop a new methodology for highly-efficient wet-chemical synthesis of nanomaterials.

In contrast to the conventional approaches, in which only slow and near-equilibrium heating is believed to be favorable for controllable wet-chemical synthesis, we propose an ultrafast yet programmable wet-interfacial Joule heating (WIJH) strategy to synthesize nanomaterials. As a proof-of-concept, the synthesis system is constructed by a thin layer of precursor solution coating on a graphene film (GF) (Fig. 1b). Joule heat generated by the GF is confined to the desired precursor layer to yield a steep ramping rate (for example,

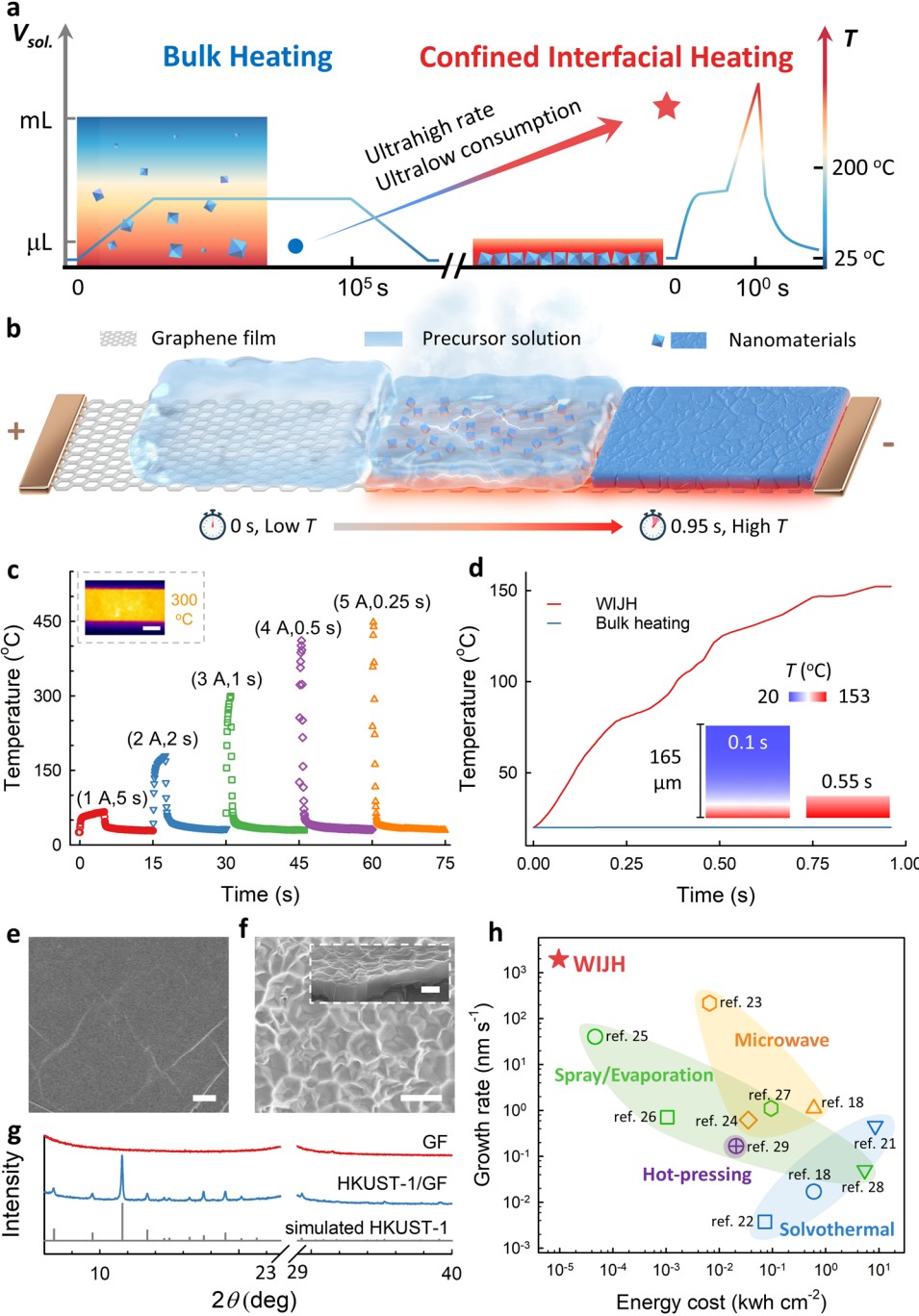

**Fig. 1 | WIJH strategy for the wet-chemical synthesis. a** Schematic of the thermal distribution and the temperature evolution of different heating strategies for wet-chemical synthesis. Note that only a heating side (from the bottom) was illustrated in the bulk-heating-based system. **b** Schematic of the WIJH system. **c** Temperature evolution of the GF with customized electrified procedures. Inset is a typical thermal image during a Joule heating process (3 A for 1 s, scale bar: 0.1 cm). **d** Estimated average temperature evolution of the solution part in different heating approaches. The insets are the typical thermal distribution images of the solution part during a WIJH process. SEM images of (**e**) GF and (**f**) HKUST−1/GF. Scale bar: 2 μm.The inset of (**f**) is the cross-sectional SEM image. Scale bar: 1 μm. **g** XRD patterns of GF, HKUST-1/GF, and simulated HKUST-1. **h** Comparison of the synthesis efficiency (growth rate and energy cost) for MOFs on the substrates by typical heating-based approaches. Source data of (**c**, **d** and **g**) are provided in the Source Data file.

around 300 °C s$^{-1}$ in Fig. 1a). Meanwhile, the precursors are further concentrated, accompanied by the instantaneous evaporation of the solvent. Thanks to the synergy between enhanced temperature (Joule heating) and concentration (evaporation), two essential elements of reaction kinetics, WIJH accelerated the reaction exponentially and realized the syntheses of various materials in (sub-)seconds, including metal-organic frameworks (MOFs), covalent-organic framework (COF), metal, metal oxide, and metal sulfide. Typically, a dense and fusing MOF (HKUST-1 as the model) film was obtained as fast as 0.25 s, in which the growth rate reached -1.97 μm s$^{-1}$ (up to 10$^5$ times), while the energy cost decreased to $9.55 \times 10^{-6}$ kWh cm$^{-2}$ (up to 10$^{-6}$ times), in comparison with conventional bulk-heating-based methods. Significantly, the WIJH could modulate the nucleation and growth of the crystals on the GF via programming the electrified procedure, thereby enabling precise control of the products' amount, size, and morphology. This is hard to be realized in conventional methods, particularly in a short second-scale time. As an application demonstration, the obtained HKUST-1/GF enables the interfacial-heating-controlled capture and liberation towards CO$_2$ with superior efficiency. This work presents an exciting breakthrough in Joule heating technology for the superefficient and programmable solvent-involved wet-chemical synthesis, based on the synergy of temperature and concentration effects in the confined interfacial heating mechanism. It differs from recent advances achieved by solid-state systems, in which only the temperature factor accounts for the trigger, acceleration, and control of the synthesis.

## Results

### WIJH for the ultrafast synthesis of prototypical HKUST-1

The ultrafast electrothermal-responsive substrate and the proximal precursor solution are two essential parts to conduct WIJH synthesis. A stable and conductive GF was selected as the internal heat source and the reaction interface (Supplementary Discussion 1). It is facile to program the heat output of the GF via programming the electrified procedures in terms of current intensity, duration time, trigger (on), stop (off), and on-off cycles, as displayed in Fig. 1c. Unlike conventional heaters with poor heat transfer and large thermal inertia, GF exhibits a high ramping rate of -1700 °C s$^{-1}$ and cooling rate of 410 °C s$^{-1}$, owing to the relatively low heat capacity and high interfacial heat transfer rate[15]. Besides, the high thermal conductivity (approximately 1400 W m$^{-1}$ K$^{-1}$)[16,17] of the GF ensures a uniform thermal distribution on the interface (inset of Fig. 1c).

On the other hand, a thin layer of precursor solution was elaborately designed. Benefiting from the good wettability of the GF (Supplementary Discussion 2), a liquid film with a high area-to-height ratio formed spontaneously on the GF surface, which could rapidly accept and confine the heat around the interface to steeply increase the solution temperature. Typically, when 2 μL of the equivalent mixture of water, ethanol (EtOH), and N,N-dimethylformamide (DMF, typical solvent for wet-chemical synthesis) was spread on the GF, it formed a thin liquid film with a height of around 165 μm and an area of 20 mm$^2$ (Supplementary Fig. 3). Followed by supplying a d.c. pulse of 3 A to the GF, the solution temperature instantaneously increased to its maximal value of 153 °C (Fig. 1d), and the system was quickly cooled down to 60 °C in 0.35 s after cutting off the current (Supplementary Fig. 4). The rapid ramping and cooling could timely trigger and stop the synthesis, respectively. Moreover, in the open system, the solvent was evaporated instantaneously during the WIJH process, as proved by a rapid fading rate of around 173.7 μm s$^{-1}$ of the liquid film (Supplementary Discussion 3). This not only further contributes to a higher heating rate, but also remarkably elevates the precursor concentration, providing another critical factor to accelerate and control the reaction. Besides, evaporation gifts the synthesis with a high utilization efficiency of the precursor, which sharply reduces the dosage requirement towards the reactants (see discussion below). In contrast, when

simulating bulk heating of 2 mL solution, it exhibits a low ramping rate of 0.16 °C h$^{-1}$, as well as slow evaporation (a low height decrease ratio of 13.3% in 24 h) (Supplementary Fig. 6).

To demonstrate the strategy, we begin with the WIJH synthesis of HKUST-1 (Cu$_3$(BTC)$_2$, BTC = 1,3,5-benzenetricarboxylate, an archetypical MOF), which is typically synthesized by the solvothermal reaction at 120 °C for over 12 h[18]. In our home-designed system (setup in Supplementary Fig. 7), the precursor solution is first spread onto the GF surface (85 mM Cu$^{2+}$, 55 mM H$_3$BTC, 10 μL cm$^{-2}$, normalized to the geometric area). After the WIJH process, a dried blue film appeared on the GF. Microscopically, the film consists of one layer of fusing particles (thickness around 1 μm, see discussion below) (Fig. 1e–f), which could be separated from the GF using an adhesive tape (Supplementary Fig. 8). The appearance of new peaks indexed to HKUST-1 in X-ray diffraction (XRD) patterns confirm the successful synthesis of pure crystalline HKUST-1 (Fig. 1g and Supplementary Fig. 9). In the copper elemental analysis by inductively coupled plasma-mass spectrometry (ICP-MS, Supplementary Fig. 10), negligible copper content could be detected in the washing solution, indicating the complete consumption of the precursors. The loading level of HKUST-1 on the GF was calculated to be 1.11 g m$^{-2}$. N$_2$ adsorption measurements validated that the presence of microporous HKUST-1 on GF increased the surface area and decreased the average pore size (Supplementary Fig. 11 and Supplementary Table 1). The estimated Brunauer-Emmett-Teller (BET) surface area of the coated HKUST-1 was 1054.8 m$^2$ g$^{-1}$, which is in the range of the HKUST-1 produced from other methods[18–20], further confirming the crystalline structure and high porosity.

According to the above results, the WIJH approach exhibited remarkable advantages in the ultrafast growth rate of 1.97 μm s$^{-1}$ (up to 10$^5$ times), ultralow energy cost of $9.55 \times 10^{-6}$ kWh cm$^{-2}$ (up to 10$^{-6}$ times), and ultrahigh mass production efficiency of 1660 μg s$^{-1}$ (up to 10$^4$ times) for the synthesis of MOFs on the substrates, comparing with previously-reported methods, mainly including solvothermal synthesis[18,21,22], microwave[18,23,24], spray/evaporation[25–28] and hot-pressing[29] (Fig. 1h, and detailed comparison in Supplementary Table 2). Generally, in conventional bulk-heating-based systems, the input energy is mainly used to heat up a mass of bulk solution. Both the poor heat generation/transfer capability of traditional heating accessories, as well as the high heat capacity of the solution limit the ramping rate and increase the energy consumption of the reaction system. The consumption in bulk is high but unnecessary for the synthesis on the substrates. In contrast, through rapid and direct heat generation and transfer, the WIJH approach achieves highly-efficient utilization of heat to accelerate the reaction on the substrates.

### WIJH mechanism for ultrafast wet-chemical synthesis

High temperature and high concentration can improve the reaction rate by accelerating the collisions between reactants. According to classic nucleation and growth theories, the nucleation rate is exponentially increased with temperature and the supersaturation ($S$, related to concentration)[30]:

$$\frac{dN}{dt} = A \exp\left(-\frac{16\pi\gamma^3 v^2}{3k^3 T^3 (\ln S)^2}\right) \qquad (1)$$

where $N$ is the number of nuclei, $A$ is the pre-exponential factor, $\gamma$ is surface free energy, $v$ is the molar volume, and $k$ is Boltzmann's constant.

As a result of the confined interfacial heating in the WIJH, we speculate both high temperature and the concentrated precursors synergistically accelerate the reaction. To testify the mechanism, we analyzed the evolution in a typical WIJH process of 3 A for 0.95 s, in terms of temperature (Fig. 2a), concentration (related to the decrease ratio of the height of the liquid film, Fig. 2b), and crystalline information (Fig. 2d, e, and Supplementary Figs. 9 and 12). According to the

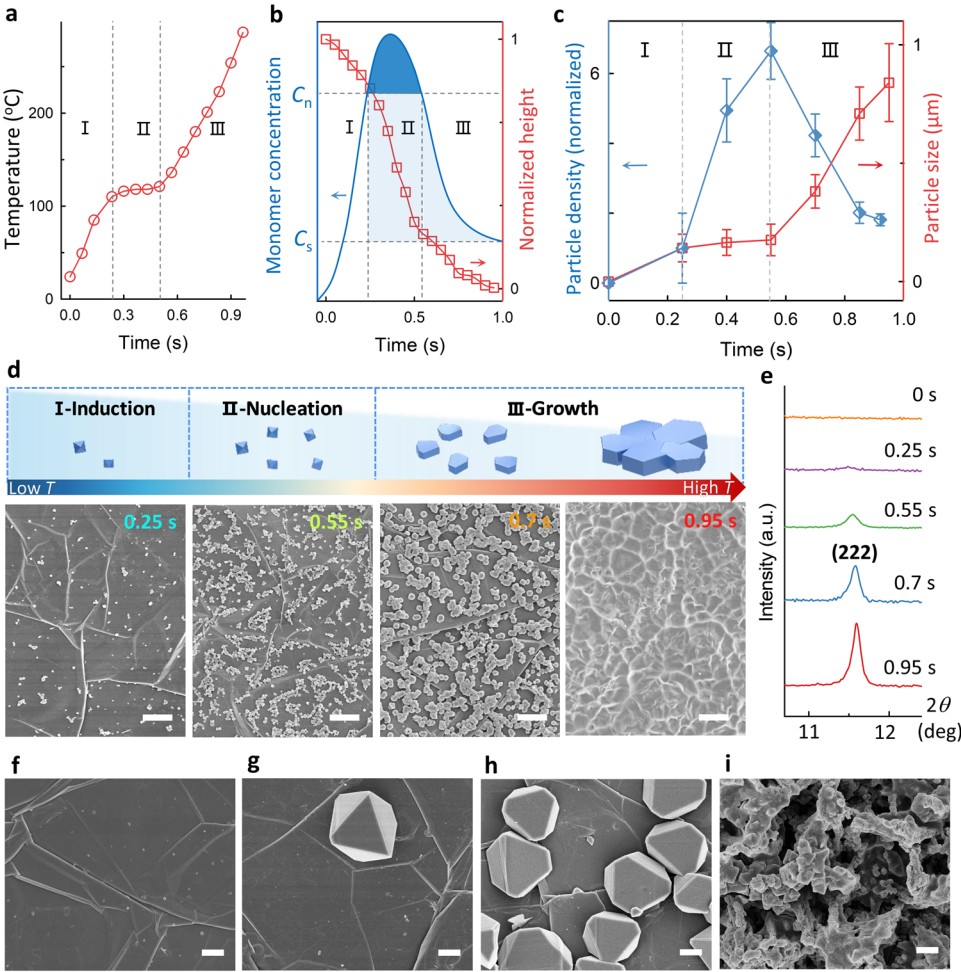

**Fig. 2 | WIJH mechanism for the ultrafast synthesis through three portions of I-induction, II-nucleation, and III-growth. a** Temperature evolution of the system consists of GF and the liquid film. **b** Schematic of the LaMer model (blue part, where $c_n$ and $c_s$ represent the critical and the saturation concentration of nucleation, respectively), and the normalized height of the liquid film (red part, which was obtained by the simulation). **c** A conclusive formation diagram of the nucleation and growth, according to the statistics of the particle densities and sizes from the SEM images. The particle densities were normalized to the value of the sample prepared within 0.25 s. Error bars represent the standard deviations of measurement from three samples. **d** The schematic (the upper) and the corresponding SEM images (the bottom) of the structural evolution from regular octahedron to truncated octahedron, and finally, the fusing film. Scale bar: 2 μm. **e** XRD evolution of HKUST-1/GF. SEM images of the products obtained in comparison experiments of (**f**) room temperature synthesis (12 h), (**g**) solvothermal synthesis (1 h), (**h**) bulk Joule heating (10 min), and (**i**) evaporation (1 h). Scale bar: 1 μm. Source data of (**a**, **b**, **c**, and **e**) are provided in the Source Data file.

statistical results (Fig. 2c), the crystallization process can be described as three elementary portions: I-induction (0–0.25 s), II-nucleation (0.25–0.55 s), and III-growth (0.55–0.95 s).

In I, the temperature rapidly increases from 23.9 °C to 115 °C due to the rapid and powerful Joule heating of GF (Fig. 2a). The high temperature could induce the initial nucleation (Supplementary Discussion 4). Meanwhile, the concentration of reactive species (monomers) also increases along with relatively-slow solvent evaporation, synergistically inducing early nucleation with a few small nuclei (Fig. 2d –0.25 s). As temperature achieves and remains at a high level of approximately 120 °C in II, the evaporation proceeds dramatically, resulting in a steep decrease of the normalized height from 0.81 to 0.22 (region II in Fig. 2b). This could markedly elevate the concentration to exceed the critical concentration of nucleation ($c_n$) in LaMer diagram, and thus, the nucleation event bursts. Consequently, lots of octahedron crystals with an average size around 175 nm were found on the GF (Fig. 2d –0.55 s), leading to a sudden rise of the normalized particle density from 1 at 0.25 s to (6.6 ± 0.8) particles per μm² GF at 0.55 s (region II in Fig. 2c). The concentration effect to promote the nucleation was further demonstrated by the increased average nucleation rates within the increased concentration of the precursors

(Supplementary Discussion 4). Besides, in XRD characterization, the full width at half maximum (FWHM) of HKUST-1 (222) was 0.14 at 0.55 s, confirming a good crystallinity (Fig. 2e).

Afterward, the concentration declines to a level below $c_n$ due to the rapid consumption of the precursors in II (region III in Fig. 2b), driving the crystallization into growth-dominant III. The particle size near-linearly increased to 854 ± 165 nm in the following 0.4 s (region III in Fig. 2c), in which high temperature was demonstrated to play a critical role in this stage (region III in Fig. 2a, Supplementary Discussion 4). Meanwhile, the ongoing evaporation could further compensate for the consumed precursors, synergistically enabling a high growth rate. In contrast, in the conventional sealed reaction system, the precursor concentration drops rapidly and dramatically as the reaction proceeds, slowing down the crystallization kinetics considerably[31]. Furthermore, the XRD peaks of HKUST-1 became stronger and sharper with prolonged reaction time, and the FWHM of the HKUST-1 (222) peak narrowed to 0.12° after 0.7 s, indicating a continuous crystallization process (Fig. 2e). Besides, as nucleation and growth are intense and successive, the particle size was relatively uniform. This is in sharp contrast with the broad size distribution obtained by conventional solvothermal syntheses

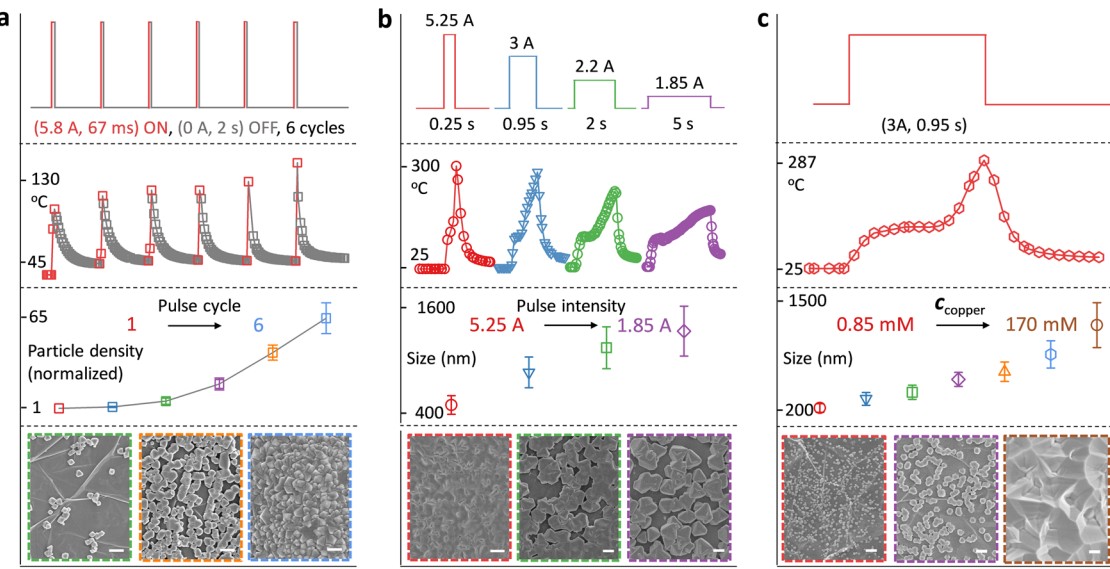

**Fig. 3 | WIJH-based programmable synthesis of HKUST-1 on the GF. a** Pulse-cycle-based modulation towards particle density. **b** Pulse-intensity-based modulation towards particle size. **c** Concentration-based modulation towards the particle size and dispersion state. From top to bottom are the electrified procedure (the current pattern); the temperature profile of the WIJH system; and statistical results of (**a**) the particle density (normalized to the value of the sample prepared within 2 pulses), (**b**) the particle size, and (**c**) particle size; and the typical SEM images of HKUST-1/GF (the border color of the image was used to mark the corresponding sample in the statistical result, scale bar: 1 μm for all). Error bars represent the standard deviations of measurement from three samples. Source data of the temperature profiles and statistical results are provided in the Source Data file.

(Supplementary Table 3), which is presumably explained by the simultaneous nucleation and growth[32].

Specifically, differing from rugged films composed of randomly stacked or intergrown particles by conventional methods[18,21,32–34], a flat and fusing film was found, which is presumably ascribed to high-temperature-induced flattening and fusion growth. In portion III of growth, the temperature gradient between the GF and the solution causes the growth difference of the formed nuclei along different directions. Notably, the growth along the horizontal direction is favorable due to the high-temperature plane of GF, and thus, truncated octahedrons appeared at 0.7 s (Fig. 2d −0.7 s). As the crystal expands in the horizontal direction rapidly, the interparticle gaps are sharply narrowed, followed by the intergrowth of neighboring crystals when meeting (Supplementary Fig. 12e −0.85 s). Finally, with the aid of the unique spatial confinement of the residual solution, particles fuse together to form a continuous thin film consisting of one-layer particles (Fig. 2d −0.95 s). The unique fusing film is expected to possess superior performance in applications that heavily rely on the degree of perfection on structure and surface, such as adsorption and separation, electronics, and sensors[35].

To further confirm the crucial role of confined interfacial heating for the ultrafast synthesis, comparison experiments were performed via monitoring the growth of HKUST-1 on the GF under different conditions, including (C1) room-temperature, 400 μL precursor solution; (C2) solvothermal synthesis at 120 °C, 400 μL; (C3) bulk Joule heating, 400 μL; (C4) WIJH, 2 μL; (C5) evaporation at room temperature and at 120 °C, 2 μL (Details in Supplementary Discussion 5). Compared with the few and small nuclei obtained by C1 (Fig. 2f), all heating-based conditions showed improved surface coverage ratios and crystallization rates (Figs. 2g–2i and Supplementary Table 3). This verifies the importance of heating to trigger and accelerate the reaction. However, with thermal dissipation by the bulk solution, the growth rates of bulk heating (-5.5 nm s⁻¹ at C2 and -21.7 nm s⁻¹ at C3) are much slower than that of WIJH (-898.9 nm s⁻¹ at C4), highlighting the crucial role of the thin liquid film for heat confinement and precursor concentration. As for the comparison of the synthesis efficiency on specific GF surfaces, WIJH realized a complete surface coverage of

100% in only 0.95 s and even 0.25 s, while bulk Joule heating at C3 exhibited a high coverage ratio of 25.8% over 6% of solvothermal condition at C2. This indicates the merit of Joule heating for the synthesis on the substrate, which benefits from the high-temperature-induced spatial controllability. In addition, the evaporation-caused concentration of the precursors also promotes crystallization. Unfortunately, rapid evaporation under oven heating led to a vast and ill-defined deposition layer at C5, in which copper hydroxide nitrate was observed besides HKUST−1 (Supplementary Fig. 21). This reflects that evaporation-caused precipitation of the precursors could impact the coordination reaction for HKUST-1. In the WIJH, the unique confined interfacial heating ensures a rapid and favorable crystallization, which can consume the concentrated precursors timely, thereby leading to pure HKUST-1.

## WIJH for the programmable synthesis on the GF

In addition to the prominent improvement in synthesis efficiency, the WIJH strategy opens up a new way to customize the wet-chemical synthesis, even in a short second-scale time (Fig. 3). This is in contrast to the general consideration of the puzzling contradiction between high rate and high controllability, and it is also hard to be realized in conventional bulk-heating-based methods. Stemming from the sensitive thermal modulation towards the reaction system by delicately programming the electrified procedure, WIJH could control the nucleation and growth on the substrate, and thus, customize the products. The particle density was adjustable via programming periodic pulses for nucleation (Fig. 3a and Supplementary Fig. 22). In a typical short pulse of 67 ms, the temperature of the system is switched from low to high (ON state) to initiate the synthesis reaction and then is quickly quenched by cutting off the pulse (OFF state). In this duration, the nucleation is allowed with the heat input, while further growth is suppressed with quick cooling. As the on-off pulses cycled, the nucleation events repeated. The normalized particle density increased from 1 particle per μm² GF after two pulses to (63.6 ± 10.9) particles per μm² GF after six pulses.

In another pulse-intensity-based modulation, the particle size was adjustable (Fig. 3b). Generally, a rapid and intense nucleation event

leads to small particles by depleting the precursors, while larger particles are obtained when growth is favorable[36]. In the WIJH, under a low-intensity pulse with a low temperature for a long duration time, finite nuclei were produced in the nucleation portion, leaving sufficient reactive species unreacted. Afterward, the growth proceeded and became preferable over time. As a result, larger and dispersed particles were obtained. As the pulse intensity increased, the size of the particle reduced from $1281 \pm 267$ nm to $491 \pm 100$ nm.

Crystallization requires high precursor concentration to overcome $c_n$. Particularly for the wet-chemical synthesis on the substrate, the concentration is among sub-mole per liter scale, due to the inherent difficulty of the heterogeneous nucleation on inert substrates and the competitive reactions in the bulk solution. Benefiting from the high utilization efficiency of the precursors, we can obtain HKUST-1 nanoparticles on the GF using an initial concentration as low as 0.85 mM copper salt, which is two hundred times lower than conventional methods. In contrast, no particle was found on the GF or in the solution within the same conditions by the solvothermal method (Supplementary Fig. 23). Moreover, by adjusting the initial concentration, HKUST-1 presents as monodispersed nanoparticles of $207 \pm 52$ nm or the intergrown film composed of particles around $1208 \pm 271$ nm (Fig. 3c and Supplementary Figs. 24 and 25). Smaller nanoparticles (with the size of $54 \pm 11$ nm) and larger microparticles of HKUST-1 ($1.65 \pm 0.2 \mu$m) could be obtained via the simultaneous control on the electrified procedure and the initial concentration, achieving the customization of the products within a wide size range (Supplementary Fig. 26).

### The universality of the WIJH approach

Through applying different WIJH procedures, we have successfully synthesized other nanomaterials in second-scale highly-efficient manners, including other MOFs (ZIF-8 (Fig. 4a)[37], MIL-88A(Fe) (Supplementary Fig. 27)[38], and Tb-BTC (Supplementary Fig. 28)[39]), COF via covalent condensation (TAPB-DMTA (TPB-DMTP-COF), Fig. 4b and Supplementary Fig. 29)[1], metal via reduction (Au nanoparticles, Fig. 4c)[40], metal oxide via redox ($MnO_2$, Fig. 4d)[41], and metal sulfide via hydrothermal decomposition (CdS, Fig. 4e)[42]. Therefore, we believe that WIJH could be a versatile and highly-efficient strategy for fabricating various nanomaterials.

In addition, the WIJH strategy was further extended to the syntheses on other substrates with different components and structures. After depositing $Al_2O_3$ layer on the GF, we achieved a HKUST-1 film on the $Al_2O_3$ in a 0.95 s WIJH process (Fig. 4f). Beyond two-dimensional flat substrates, rapid syntheses (<10 s) of HKUST−1 were also realized on one-dimensional carbon fiber (CC, Fig. 4g)[43,44] and three-dimensional graphene aerogel (GA, Fig. 4h)[45–47].

Moreover, with the aid of a roll-to-roll Joule-heating setup, the approach can be scaled up easily for the continuous and mass fabrication of nanomaterials, providing potential opportunities for industrial-scale production at atmospheric pressure (Supplementary Fig. 30). For one batch of the fabrication of HKUST−1/GF, both fusing HKUST-1 film and dispersed nanoparticles could be obtained through altering the electrified procedures, giving the production rates around $14.7$ cm$^2$ s$^{-1}$ and $25$ cm$^2$ s$^{-1}$, respectively (Supplementary Figs. 30b and 30c). The samples from different sites along the length direction of the upscaled HKUST-1/GF exhibited similar morphologies, indicating a relatively uniform distribution of the nanomaterials across the substrate in a long range. MOF loading can be further scaled and easily adjusted through a repeating layer-by-layer fashion, producing intergrown multi-layer HKUST-1 coatings on the GF (Supplementary Fig. 30d). From the first to five cycles, the mass rises from ($1.13 \pm 0.33$) to ($4.67 \pm 0.38$) g m$^{-2}$ (Supplementary Fig. 30f). Besides, the HKUST-1/GF can also be obtained with a simple circuit supplied by four AA batteries (Supplementary Fig. 31), making it potential for further portable applications.

### IJH-controlled capture and liberation towards $CO_2$

Beyond material synthesis, heating also participates in various critical thermochemical processes, e.g., temperature swing adsorption (TSA) of gas. TSA is considered an energy-saving route for gas liberation compared to other swing processes driven by pressure and vacuum. Nevertheless, in the case of the release from MOFs, their low thermal conductivities hinder the process, thereby strongly decreasing the efficiency[48,49]. Within a powerful heat output and the direct transfer from GF to the loaded MOFs, our interfacial Joule heating (IJH) strategy could control the adsorption and desorption of gas in a highly-efficient manner.

As a proof-of-principle application, we tested the IJH-controlled $CO_2$ capture and liberation (utilization) based on the obtained HKUST-1/GF (Fig. 5a), which was expected to help to tackle global warming and growing energy demand[50]. As displayed in Fig. 5b, the capture capacities of GF, and HKUST-1/GF that were prepared by solvothermal and the WIJH methods were 2.8, 5.73, and 14.56 cm$^3$ g$^{-1}$ at 298 K and 1 bar, respectively. The capacities of HKUST-1$_{WIJH}$ and HKUST-1$_{solvothermal}$ were correspondingly calculated to be 3.95 and 3.78 mmol g$^{-1}$, which are comparable with the values reported by other works[51,52]. HKUST-1 could strongly adsorb $CO_2$ with the electrostatic interactions between open $Cu^{2+}$ sites and $CO_2$ molecules in an end-on fashion ($O=C=O\cdots Cu^{2+}$)[53], while the low capacity of HKUST-1/GF by solvothermal synthesis is mainly ascribed to the few and nonuniform loading of HKUST-1 on the GF (Supplementary Fig. 32a). Besides, the IJH-based programmable capture and liberation were demonstrated via applying the in-situ customized pulses during the adsorption process (Supplementary Fig. 33). The adsorption is an exothermic physical process. The kinetic energies of the adsorbed $CO_2$ and HKUST-1 are higher under Joule heating. This causes a corresponding increase in their interaction at the interface, thereby reducing the effective area of HKUST-1 available for adsorption[54]. Therefore, the uptake would decrease, and different amounts of the adsorbed $CO_2$ could be liberated by applying pulses with different densities and duration times (Fig. 5c and Supplementary Fig. 32b). It was found that a low energy cost of $3.72 \times 10^{-4}$ kWh was required to release 1 cm$^3$ $CO_2$. Moreover, the IJH-controlled adsorption and desorption exhibited excellent reversibility and stability, as proved by the barely changed efficiency even after 20 cycles (Fig. 5d).

## Discussion

In summary, we present a creative WIJH route to synthesize nanomaterials with super-efficiency, programmability, versatility, and the potential for scalability. In the WIJH system, a mechanism of confined interfacial heating was proposed to manipulate the synthesis, in which temperature and concentration were simultaneously modulated, thereby achieving rapid and controllable nucleation and growth. For the synthesis of MOFs on the substrates, the growth rate of 1.97 μm s$^{-1}$ is up to 5 orders of magnitude faster, while the energy cost of $9.55 \times 10^{-6}$ kWh per square centimeter substrate is 6 orders of magnitude lower than the previously-reported heating-based methods. Typical nanomaterials, including MOFs, COF, metal, metal oxide, and metal sulfide, were successfully obtained in seconds. Further, through heating-manipulated reaction thermodynamics and kinetics, the synthesis is programmable in the product's amount, size, and morphology. The obtained HKUST-1/GF presented IJH-controlled and highly-efficient capture and liberation towards $CO_2$. We believe that WIJH synthesis should be a powerful methodology and a tool in scientific research and practical application for both nanomaterials and wet chemistry.

## Methods

### Chemicals and materials

All the chemicals were of analytical grade and used as received without further purification. Copper nitrate hemi(pentahydrate)

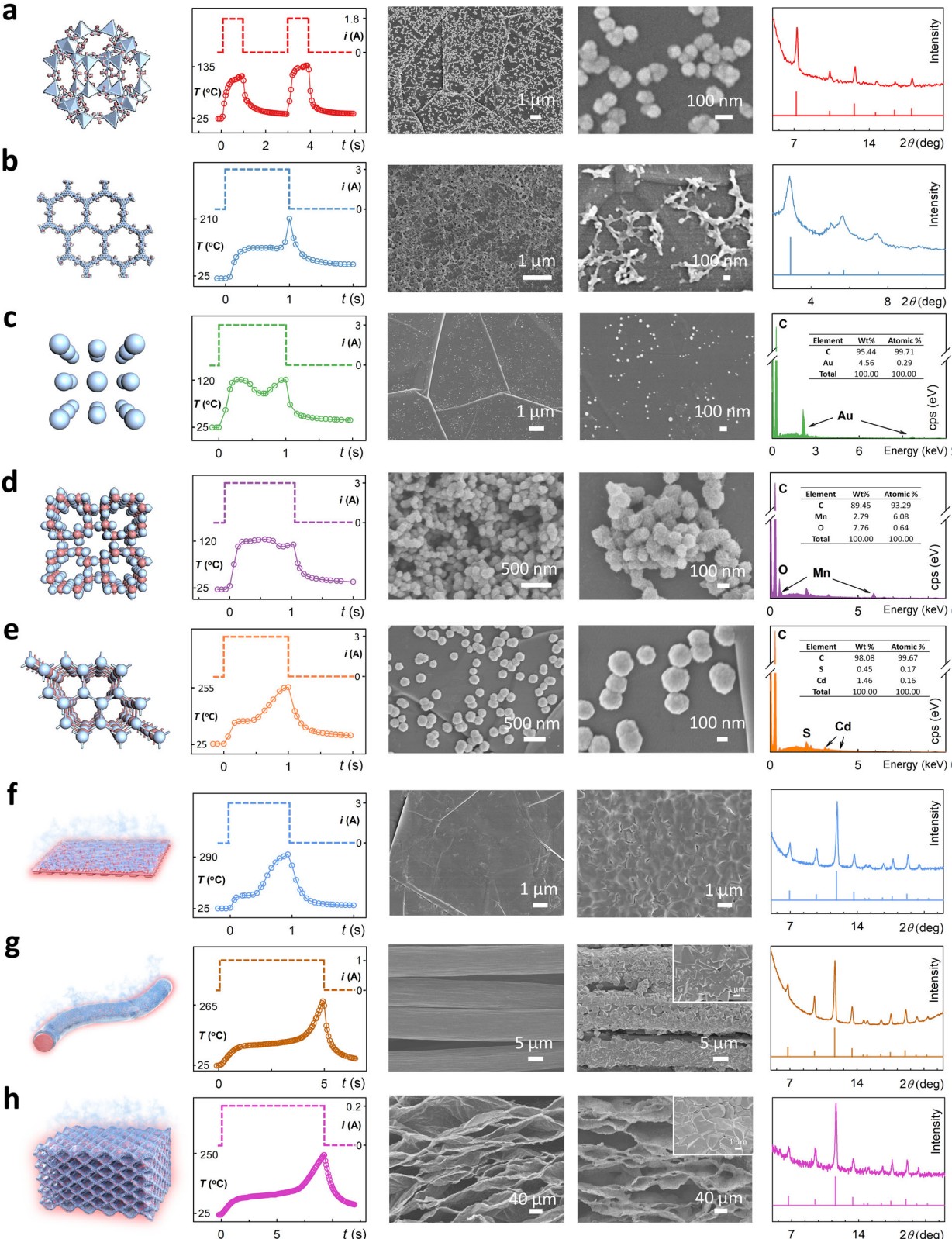

**Fig. 4 | The universality of the WIJH strategy.** Syntheses and characterizations of (**a**) ZIF-8/GF, (**b**) TAPB-DMTA/GF, (**c**) Au/GF, (**d**) MnO₂/GF, (**e**) CdS/GF, (**f**) HKUST-1/Al₂O₃/GF, (**g**) HKUST−1/CC, (**h**) HKUST−1/GA. In (**a**–**h**), from left to right are the schematic diagrams (crystal structure in (**a**–**e**) and approach in (**f**–**h**)); the electrified procedures (the top) and the temperature profiles (the bottom); SEM images with low and high magnification scales; and characterizations (XRD patterns of experimental samples (the top) and simulated nanomaterials (the bottom) in (**a**–**b**) and (**f**–**h**), elemental mapping results in (**c**–**e**)). Source data of the temperature profiles, XRD patterns, and elemental mapping are provided in the Source Data file.

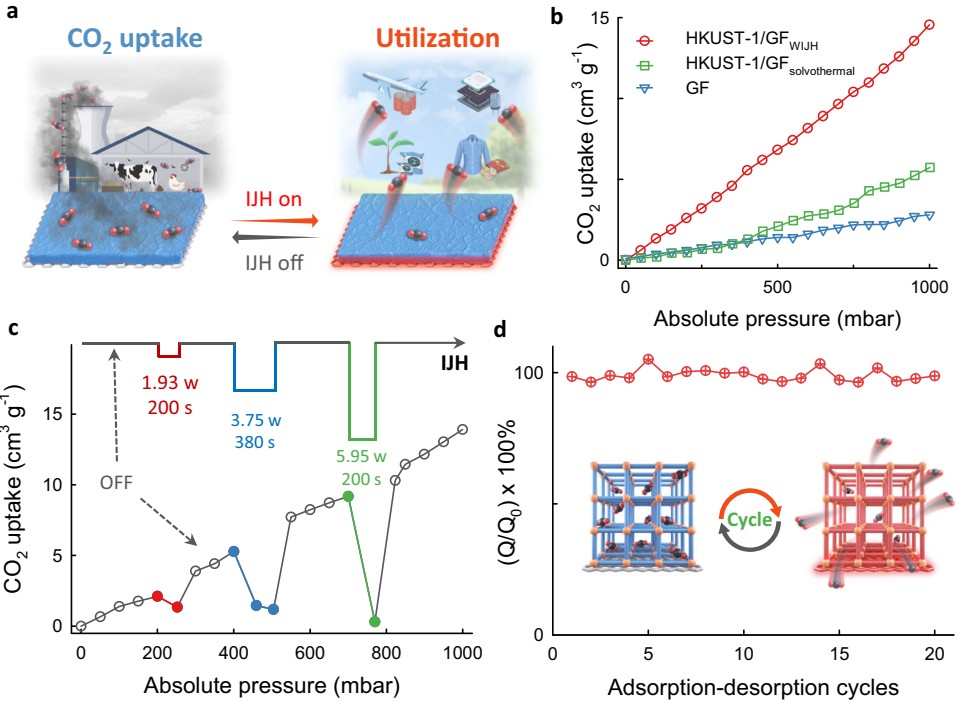

**Fig. 5 | IJH-controlled capture and liberation towards CO₂ based on HKUST-1/GF. a** Schematic. **b** CO₂ adsorption isotherms at 25 °C. **c** CO₂ adsorption isotherm of HKUST−1/GF under programmed IJH processes. **d** Stability of adsorption-desorption cycles. The inset is the schematic of the IJH-controlled reversible capture and liberation. Source data of (**b**, **c**, and **d**) are provided in the Source Data file.

(Cu(NO₃)₂·2.5H₂O), zinc nitrate hexahydrate (Zn(NO₃)₂·6H₂O), trimesic acid (H₃BTC), 2-methylimidazole, fumaric acid were purchased from Sigma-Aldrich. Chloroauric acid (HAuCl₄), manganese sulfate monohydrate (MnSO₄·H₂O), potassium permanganate (KMnO₄), iron chloride hexahydrate (FeCl₃·6H₂O), trisodium citrate dihydrate, thiourea, hydrazine hydrate (N₂H₄), acetic acid, DMF, EtOH, methanol, acetone, and dichloromethane were purchased from Sinopharm Chemical Reagent Co. Ltd. Cadmium nitrate tetrahydrate (Cd(NO₃)₂·4H₂O), terbium nitrate hexahydrate (Tb(NO₃)₂·6H₂O) were obtained from Aladdin Industrial Corporation. 1,3,5-Tris(4-aminophenyl)benzene (TAPB) and 2,5-dimethoxybenzene-1,4-dicarboxaldehyde (DMTA) were purchased from Chemsoon Co. Ltd. Graphene oxide (GO) was produced by Hangzhou Gaoxi Technology Co. Ltd. (www.gaoxitech.com). CC was purchased from CeTech Co., Ltd.

### Fabrication of graphene macroscopic materials as the Joule heating sources

Graphene macroscopic materials were fabricated according to the previous works after optimizations[55]. Briefly, GF[16] was fabricated by casting debris-free and giant GO dispersions (concentration of 6 mg mL⁻¹), followed by thermal annealing at 1300 °C for 2 h and 2800 °C for 2 h. For hydrophilic treatment, air plasma (Harrick Plasma, PlasmaFlo PDC-FMG−2) was applied to the surface of GF. GA[45] was fabricated via the hydroplastic foaming method. Free-standing GO film (thickness of 30 μm) was immersed in the N₂H₄/H₂O solution (30%) for 1 h. After drying, GA was further annealed at 1600 °C for 1 h.

### General procedure for the WIJH synthesis

The WIJH synthesis was conducted with a custom-built setup at atmospheric pressure (schematic diagram in Supplementary Fig. 7). A free-standing conductive substrate was fixed on a trench formed by a glass holder, and was connected with a d.c. power source (ITECH, IT6431) through silver glue and copper foil. Several microliters of the precursor solution were carefully spread on the reaction area of the substrate (the suspended area on the trench) for wetting. Immediately,

the wet precursor-loaded substrate was subjected to a customized d.c. pulse by the power source (current output mode, basic parameters of current, time, resistance of 0 Ω, and the maximum output voltage of 15 V). Note that the temperature of the WIJH system should be less than the decomposition temperature of the nanomaterials (Details in Supplementary Discussion 6). Finally, the product was integrally peeled and washed, followed by drying.

### Synthesis of HKUST-1/GF

In a typical WIJH synthesis of HKUTS-1 on the GF, the precursors of 85 mM Cu(NO₃)₂ and 55 mM H₃BTC were firstly dissolved in a mixture of H₂O-EtOH-DMF (volume ratio of 1:1:1). 2 μL of the precursor solution was spread on the reaction area of the GF (1 cm × 0.2 cm). Then, the film was immediately subjected to a d.c. pulse of 3 A for 0.95 s. Afterward, the resulting film was peeled from the setup and washed with DMF and EtOH, followed by drying at 80 °C for 12 h. For the exfoliation of HKUST−1 from the GF, an adhesive tape was first pressed onto the HKUST-1 layer and removed from the GF surface in one quick swoop. Other samples of HKUST−1/GF for specific experiments were prepared by varying the initial concentrations of the precursors (with the same 3:2 molar ratio of Cu(NO₃)₂ and H₃BTC) and the electrified procedure in terms of current intensity, duration time, and cycle times.

### Synthesis of MIL-88A(Fe)/GF

The precursors of 40 mM FeCl₃·6H₂O and 40 mM fumaric acid were firstly dissolved in a mixture of H₂O-EtOH-DMF (volume ratio of 1:1:2). 1.8 μL of the solution was spread on the GF surface. Then, the precursor-loaded GF was subjected to a d.c. pulse of 3 A for 0.95 s, followed by peeling, washing with DMF and EtOH, and drying at 80 °C for 12 h.

### Synthesis of ZIF-8/GF

The precursors of 80 mM Zn(NO₃)₂·6H₂O and 80 mM 2-methylimidazole were first dissolved in a mixture of H₂O-methanol-DMF (volume ratio of 1:1:2). 1.6 μL of the solution was spread on the GF surface. The precursor-loaded GF was then subjected to a series of

successive d.c. pulses of 1.85 A for 0.95 s (ON state), 0 A for 2 s (OFF state); 1.85 A for 0.95 s (ON state), and 0 A (OFF state) to stop the synthesis. Finally, the resulting film was washed with methanol, followed by drying at 60 °C for 12 h.

### Synthesis of Tb-BTC/GF

The precursors of 1.5 mM Tb(NO$_3$)$_2$·6H$_2$O and 0.5 mM H$_3$BTC were first dissolved in a mixture of H$_2$O-DMF (volume ratio of 2:3). 2 μL of the solution was spread on the GF surface. The precursor-loaded GF was then subjected to a series of successive d.c. pulses of 2.25 A for 0.95 s (ON state), 0 A for 2 s (OFF state); 2.25 A for 0.95 s (ON state), 0 A for 2 s (OFF state); 1.25 A for 0.95 s (ON state), 0 A for 2 s (OFF state); 1.25 A for 0.95 s (ON state), 0 A (OFF state) to stop the synthesis. Finally, the resulting film was washed with DMF and water, followed by drying at 60 °C for 12 h.

### Synthesis of TAPB-DMTA/GF

The precursors of 40 mM TAPB and 60 mM DMTA were first dissolved in 6 M acetic acid aqueous solution. 1.25 μL of the solution was spread on the hydrophilic GF. Then, the precursor-loaded GF was subjected to a d.c. pulse of 3 A for 0.95 s. The resulting film was washed in sequence with acetone, dichloromethane, and methanol, followed by a 24 h Soxhlet extraction with methanol. Finally, the sample was dried under a high vacuum for 24 h.

### Synthesis of Au/GF

The precursor solution was prepared by mixing 2 μL of 1% HAuCl$_4$, 15 μL of 1% trisodium citrate, and 107 μL of water. 1.8 μL of the solution was spread on the hydrophilic GF. Then, the precursor-loaded GF was subjected to a d.c. pulse of 3 A for 0.95 s. Finally, the resulting film was washed with water, followed by drying at 60 °C for 12 h.

### Synthesis of MnO$_2$/GF

The precursors of 15 mM MnSO$_4$ and 40 mM KMnO$_4$ were dissolved in water. 1.8 μL of the solution was spread on the hydrophilic GF. Then, the precursor-loaded GF was subjected to a d.c. pulse of 3 A for 0.95 s. Finally, the resulting film was washed with water, followed by drying at 60 °C for 12 h.

### Synthesis of CdS/GF

The precursors of 40 mM Cd(NO$_3$)$_2$ and 120 mM thiourea were first dissolved in water. 0.75 μL of the solution was spread on the hydrophilic GF. The precursor-loaded GF was then subjected to a d.c. pulse of 3 A for 0.95 s. Finally, the resulting film was washed with water, followed by drying at 60 °C for 12 h.

### Synthesis of HKUST-1 on Al$_2$O$_3$/GF

Al$_2$O$_3$/GF was firstly prepared by depositing Al$_2$O$_3$ layer (about 50 nm) on the GF (Kurt j.Lesker, 75PRO-line). 2 μL of the precursor solution of 85 mM Cu(NO$_3$)$_2$·2.5H$_2$O and 55 mM H$_3$BTC was spread on the Al$_2$O$_3$ layer upon GF surface, and then, the film was subjected to a d.c. pulse of 3 A for 0.95 s. Afterward, the resulting film was integrally peeled and washed with DMF and EtOH, followed by drying at 80 °C for 12 h.

### Synthesis of HKUST-1/CC

CC was cut into a strip with a length of 2.5 cm and a width of 0.5 cm for the WIJH synthesis. After assembling into the setup, 15 uL of the precursor solution of 170 mM Cu(NO$_3$)$_2$ and 110 mM H$_3$BTC was added to the CC. Immediately, the precursor-loaded CC was subjected to a d.c. pulse of 1.05 A for 5 s for the growth of HKUST-1. Finally, the sample was washed with DMF and EtOH, followed by drying at 80 °C for 12 h.

### Synthesis of HKUST-1/GA

GA was cut into a cube with a length of 1.5 cm, a width of 0.2 cm, and a height of 0.2 cm for the WIJH synthesis. After assembling into the setup, 10 uL of the precursor solution of 170 mM Cu(NO$_3$)$_2$ and 110 mM H$_3$BTC was added from the cross section of the GA. The precursor-loaded GA was subjected to a d.c. pulse of 0.2 A for 9.25 s for the growth of HKUST-1. Afterward, the resulting sample was peeled and washed with DMF and EtOH, followed by drying at 80 °C for 12 h.

### Roll-to-roll continuous production of HKUST-1/GF

The continuous production was demonstrated on a roll-to-roll Joule-heating fabrication system according to the previous report[56]. The system mainly consists of two pairs of parallel graphite electrodes (interval distance of 7 cm), two controllable micromotors to roll the electrodes with a constant speed (e.g., 10 rpm), and a d.c. power source (ITECH, IT65220) connecting with the electrodes via two electric brushes (Supplementary Fig. 30a). A continuous GF strip with a width of 2 cm was clamped and passed through the rotating electrodes. The precursor solution of HKUST-1 (85 mM Cu(NO$_3$)$_2$ and 55 mM H$_3$BTC) was spread on the GF to form a thin-layer liquid film (around 10 μL cm$^{-2}$, on the area between the electrodes). Afterward, an instantaneous current flow (24 A, 6 V) was applied to the electrodes to conduct the WIJH synthesis. Finally, the film was collected, followed by washing and drying. For the layer-by-layer fashion, the GF was cycled into the area between two electrodes by the micromotors. A series of repeating procedures of adding the precursor solution to the same region and the WIJH synthesis were conducted successively until the fabrication was completed.

### ICP-MS measurement

ICP-MS was conducted on a spectrometer (PerkinElmer, NexION 300X) for the analysis of copper elements. To calculate the residue of copper ions, the freshly prepared blue film was statically soaked in 0.5 mL DMF for 10 min. After centrifugation, 20 μL of the supernatant was diluted into 1 mL using H$_2$O for ICP-MS test. To determine the loading of HKUST-1 on the GF, HKUST-1/GF was firstly dried at 100 °C in a vacuum oven for 12 h, and was quickly weighed and cut into tiny pieces. The sample was digested in 1 mL boiled HNO$_3$ for 2 h, and then was diluted by H$_2$O for ICP-MS test.

The mass loading (L$_m$) and the areal loading (L$_s$) are calculated as following Eqs. (2) and (3), respectively:

$$L_m = \frac{c \times V}{m \times w_{Cu}} \times 100\% \tag{2}$$

$$L_s = \frac{c \times V}{w_{Cu} \times S} \times 100\% \tag{3}$$

where $c$ is the concentration of copper in the diluted acid solution measured by ICP-MS, mg/L. $V$ is the volume of diluted digestion solution, L. $m$ is the mass of HKUST-1/GF, mg. $w_{Cu}$ is the mass percentage of copper for HKUST-1, 31.7%. $S$ is the geometric area of GF coating with HKUST-1.

### IJH-controlled adsorption/desorption towards CO$_2$

CO$_2$ adsorption/desorption measurement was conducted on the analyzer (MicrotracBEL, BELSORP-max II). CO$_2$ adsorption isotherms were routinely performed at 25 °C in a water bath without IJH process, after activating the samples at 120 °C for 6 h. The capture capacity ($Q$) of HKUST-1 on the GF towards CO$_2$ was estimated as the following Eq. (4):

$$Q_{HKUST-1}(\text{mmol g}^{-1}) = \frac{Q_{HKUST-1/GF} \times m_{HKUST-1/GF} - Q_{GF} \times m_{GF}}{m_{HKUST-1}} \tag{4}$$

where $m$ is the mass of the corresponding sample, which is determined by weighing.

IJH-controlled adsorption/desorption measurement was also conducted on the isothermal adsorption process, during which IJH was

in-situ applied to a HKUST-1/GF strip. The IJH setup consists of a power source and a custom-designed sample tube sealed by silica gel (Schematic diagram in Supplementary Fig. 33). The stability was calculated via comparing the uptake before and after an IJH process.

## Simulation of the WIJH process

The decrease in the height of the liquid film and the thermal distribution during the WIJH process were simulated with the COMSOL software. The model considers the means of heat transfer by conduction and convection. The height of the liquid film in Fig. 2b was normalized to the initial one of 165 μm. Details:

Heating condition: the temperature of the GF was obtained based on the experimental measurements by the infrared thermometer (Supplementary Fig. 5b), and the temperature evolution versus time was set as the heat input. The initial temperature of the liquid was set at 20 °C;

Liquid component: the equivalent mixture of $H_2O$, EtOH, and DMF;

Geometry: Length = 1 cm, the height of the liquid film: 165 μm (corresponding to 2 μL liquid) or 200 mm (2 mL liquid);

Thermal conductivity ($W\,m^{-1}\,K^{-1}$): GF = 1400, $H_2O$ = 0.61, EtOH = 0.18, DMF = 0.18, the liquid mixture = 0.32;

Specific heat capacity ($J\,kg^{-1}\,K^{-1}$): GF = 0.76, $H_2O$ = 4200, EtOH = 2560, DMF = 2500, the liquid mixture = 3086.7;

Density ($kg\,m^{-3}$): GF = 1800, $H_2O$ = 1000, EtOH = 789, DMF = 945, the liquid mixture = 911.3;

Boiling point (°C): $H_2O$ = 100, EtOH = 78, DMF = 153.

## Characterizations

The temperature evolution and thermal image during the WIJH process were in-situ recorded in a top view by a high-speed infrared thermometer (Teledyne FLIR, T630sc). Raman spectra were recorded on an inVia-Reflex Raman microscope (Renishaw) with a 532 nm laser source. Thermogravimetric analysis was performed on the STARe System (Mettler-Toledo, TGA2) from 30 °C with a heating rate of 10 °C min⁻¹ in air. The contact angle was measured with a Video contact Angle analyzer (Dataphysics, OCA20). The photographs before and after the spreading of the mixture of $H_2O$-EtOH-DMF were compared to estimate the height of the thin liquid film. Nitrogen adsorption-desorption isotherms and BET surface area data were measured on Quantachrome Instruments at 77 K (surface area and pore size analyzer NOVA touch LX4). Each sample was degassed at 120 °C for 6 h in the vacuum before the test. XRD pattern study was carried out using monochromatic Cu Kα1 radiation (λ = 1.5406 Å) on an X-Ray diffractometer (Bruker, D8 ADVANCE). SEM images and energy dispersion spectrum were collected on a field emission scanning electron microscope (Hitachi, SU 8010). Fourier transform infrared spectroscopy was performed using a FT-IR spectrometer (Thermo Scientific, Nicolet iS50).

## Statistics analyses

Particle density was calculated as the statistical quantity of the product particles divided by the area of the GF in the SEM image. Particle size was recorded by the SEM image using the Analyze Particles function in ImageJ, and the error bars represented the standard deviations. The growth rate was calculated as the average size of the particle divided by the synthesis time. The given time in the electrified procedure is regarded as the synthesis time since the reaction could be stopped timely within a high cooling rate after cutting off the current.

## Data availability

All the supporting data are provided in the main text and Supplementary Information. All raw data generated during the current study are available from the corresponding authors upon request. Source data are provided with this paper.

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

## Acknowledgements

This work was supported by National Natural Science Foundation of China (Nos. 52090030 (C.G.), 52272046 (Y.J.L.), 52122301 (Z.X.), 51973191(Z.X.)), Natural Science Foundation of Zhejiang Province for Distinguished Young Scholars (Grant No. LR22C170002 (Y.C.F.) and No. LR23E020003 (Y.J.L.)), Key R&D Program of Zhejiang Province (Grant No. 2021C02062 (Y.C.F.), 2022C02024 (Y.C.F.)), Fundamental Research Funds for the Central Universities (No. 226-2023-00023 (Y.J.L.), 2021FZZX001-17 (Z.X.)), Shanxi-Zheda Institute of New Materials and Chemical Engineering (2021SZ-FR004 (Y.J.L.), 2022SZ-TD011 (C.G.), 2022SZ-TD012 (Z.X.)), China Postdoctoral Science Foundation (2022M722770 (L.Z.)) and Postdoctoral Research Program of Zhejiang Province (ZJ2022005 (L.Z.)). L. Zhang thanks the financial support from the International Research Center for X Polymers, International Campus, Zhejiang University.

## Author contributions

C.G., Y.C.F., Y.J.L., and L.Z. came up with the design concept. L.Z. collected the data and the analysis for the WIJH synthesis of HKUST-1. L.P. conducted the fabrication and characterization of the GF. Y.C.L. carried out the simulation of the WIJH process and IJH-controlled capture and liberation towards $CO_2$. X.M. performed the roll-to-roll fabrication for scalability. Y.X.S., X.Y.X., Y.X.X., and K.P. conducted experiments to explore the university of the approach. K.P., W.Z.F., N.H., and Z.X. participated in discussing the experimental design and data analysis. L.Z., L.P., Z.X., Y.B.Y., Y.J.L., Y.C.F., and C.G. collectively wrote and revised the paper, together with input from all authors. C.G., Y.C.F., and Y.J.L. supervised the project. All authors discussed the results and contributed to the final manuscript.

## Competing interests

The authors declare no competing interests.
