## [Peer Review File · Nature Communications]

Sub-second ultrafast yet programmable wet-chemical synthesisREVIEWER COMMENTS

Reviewer #1 (Remarks to the Author):

This manuscript reports a wet-interfacial Joule heating (WIJH) approach to synthesize nanomaterials in a sub-second ultrafast, programmable, and energy/reactant-saving manner. Using this method, authors also demonstrate the successful synthesis of the metal-organic 23 framework, covalent-organic framework, metal, metal oxide and sulfide in an ultralow energy cost and a high growth rate. Compared with the traditional thermal treatment approaches, the Joule heating displays the great advantages in synthesis rate and energy cost. In recent years, ultrafast synthesis of nanomaterials by Joule heat has been widely studied, and some researchers also call Joule heating as high-temperature thermal shock method or carbothermal shock. Although the authors have presented the detailed discussion about the universality of this wet-interfacial Joule heating method in material synthesis, this work shows the limited novelty and scientific significance. Thus, I didn't recommend accepting this manuscript for nature communication.

1. Considering the previous publications on the Joule heating method, the authors are required to add more recent related progresses. Moreover, the main differences and the novelty of this work should be highlighted in the introduction part.
2. Fig. 2c and 2b are hard to understand and authors should add the relevant description of the symbols in the Figures, such as the C, Cs, and h0. In addition, the detailed experiment content about how to get the h in Fig. 2b and how to get the density in Fig. 2c should be provided in manuscript or supplementary information.
3. Authors have discussed the WIJH synthesis mechanism and the difference on the growth of HKUST-1 for the various synthesis conditions, have some different found in nucleation or growth mechanism for the WIJH compared with the conventional bulk-heating-based methods or solvothermal method?
4. How did authors obtain the specific value of the normalized quantity in Fig. 3a?
5. Achieving the large-scaled synthesis using Joule heating method is significant for industrial production, thus I recommend that the authors provide more experiment content and discussion on the roll-to-roll Joule-heating continuous fabrication.
6. As the control experiment, the performance of the CO₂ capture and liberation for the HKUST-1/GF synthesized by conventional method should be added and discussed.

Reviewer #2 (Remarks to the Author):

This manuscript reports the synthesis of materials using Joule heating method. These materials are often prepared by the wet-chemistry that involves a longer synthesis time and large usage of reactants/solvent etc. By placing a thin layer of synthesis precursor on the heating medium, a carbon file, the authors claimed that the materials including MOFs, COF, metal, metal oxide and sulfide were prepared within sub-second. Overall this is a very interesting work since the synthesis method is new and the formed materials morphology is quite different from the conventional thermal synthesis. Specific comments are listed below.

- (1) The materials synthesized by this Joule heating method are simple and easy to crystallize. For example, ZIF-8, HKUST etc does not need very harsh conditions to synthesize. From this point of view, it is hard to see the obvious advantages of the Joule heating method for wet-chemistry synthesis of solid materials.
- (2) The materials' synthesis solution is quick small. During the Joule heating, the evaporation of solvent should have a significant effects on the materials crystallization and growth. The authors mentioned a bit on this effect, but I think this effect should be more obvious than what has been claimed in the manuscript. Control experiments should be done to clarify the dominating mechanism (e.g., Joule heating versus solvent evaporation) in the synthesis.

(3) In line 157, the nucleation equation, I think there should be a negative sign in the exponent. Otherwise, the dependence on Temperature and supersaturation does not seem to be correct.

(4) How did the authors get the growth rate in the synthesis? The heating and cooling is fast which needs techniques to capture these temperature profiles precisely. On the basis of this, is it possible to get the rate. How does the temperature ramp up and down affect the growth?

Reviewer #3 (Remarks to the Author):

In this work, Zhang et al. reported a wet-interfacial Joule heating approach for synthesis of nanomaterials on conductive carbon films like graphene films. A very thin liquid layer is coated on the graphene film and the Joule heating rapidly brings it to a high temperature to trigger the reaction. They synthesized various nanomaterials such as MOF, COF, metal nanoparticles, oxide nanoparticles, etc. And they demonstrated the application of the MOF films for CO₂ uptake and the application of the Joule heating method for temperature swing adsorption.

Main concerns:

1. One main concern of this method is the scalability. The reaction happens at the interface between the graphene film and the solution; thus, the solution must be a very thin layer. This seriously limits its production rate and scalability compared to the bulk solution synthesis method. As demonstrated by the authors, a typical trial only uses 2 μ L solution. Could the authors demonstrate some strategies to scale up the process? Maybe a continuous process?
2. The synthesized materials are loaded onto the graphene film substrates. Is there any appropriate method to separate the synthesized materials and the heating substrates?
3. In Supplementary Table 2, the authors compared the energy cost, time consumption, etc., for a batch between the reported method here with literature routes. However, I recommend to normalize to the mass per batch since different literature routes obtained different amounts of materials. It's unfair to compare this method for 2 μ L synthesis with other methods with, e.g., 2 mL synthesis.
4. The particle size of MOF synthesized by this method is a few hundreds of nanometer. What's the size controllability of this method? Could it be used to synthesize MOF nanoparticles (<100 nm) or large crystals?
5. The authors demonstrate the CO₂ uptake and liberation cycle by the Joule heating process (Figure 5). What is the mechanism of the Joule heating-induced liberation?
6. What is the energy cost of the Joule heating induced CO₂ desorption and how does it compare to temperature swing enabled by other thermal processes, as well as other pressure swing process?
7. Will the precursors also be evaporated during the synthesis process since the synthesis temperature is so high (>300°C)? And the organics maybe be destroyed under this high temperature.

Minor points:

1. Supplementary Fig. 5 is not discussed in the main text.
2. Line 217, the authors mentioned that the particles could fuse together to form a film consisting of monolayer MOF particles. There is no evidence that the film is "monolayer".
3. PDF reference card should be provided for all the XRD patterns.

RESPONSE TO REVIEWERS' COMMENTS

Responses to Reviewer #1:

This manuscript reports a wet-interfacial Joule heating (WIJH) approach to synthesize nanomaterials in a sub-second ultrafast, programmable, and energy/reactant-saving manner. Using this method, authors also demonstrate the successful synthesis of the metal-organic framework, covalent-organic framework, metal, metal oxide and sulfide in an ultralow energy cost and a high growth rate. Compared with the traditional thermal treatment approaches, the Joule heating displays the great advantages in synthesis rate and energy cost. In recent years, ultrafast synthesis of nanomaterials by Joule heat has been widely studied, and some researchers also call Joule heating as high-temperature thermal shock method or carbothermal shock. Although the authors have presented the detailed discussion about the universality of this wet- interfacial Joule heating method in material synthesis, this work shows the limited novelty and scientific significance. Thus, I didn't recommend accepting this manuscript for nature communication.

R: We sincerely thank the reviewer for the valuable comments and suggestions to help us improve our work. We have tried our best to address these issues and revised the manuscript point-by-point.

We truly understand the reviewer's concern about the novelty and scientific significance of the reported WIJH approach, and we would like to take this opportunity to clearly compare our work with previous significant references about the Joule-heating-based ultrafast synthesis of nanomaterials (Table R1). **Differing from recent advances achieved by solid-state reactions in solid-solid systems where only temperature affects the synthesis, our work first realized ultrafast yet programmable liquid-phase wet-chemical synthesis of nanomaterials, based on a new confined interfacial heating mechanism of the synergy between Joule heating (temperature effect) and evaporation (concentration effect).** The condition, process, and mechanism for forming nanomaterials from precursor solution are significantly different from those from solid precursors. Thermal conduction in heating liquid (solution) is far from that in heating solid reported in recent works, which is also one of the greatest hindrances in developing highly efficient and controllable wet-chemical synthesis. Moreover, the evaporation-caused concentration effect, another essential factor in reaction kinetics and thermodynamics, is absent or ignored in previous works. **How to combine temperature and concentration to accelerate and control the synthesis is significant in both science and industry, but has rarely been reported before.** WIJH realized the acceleration and control of the nucleation and growth of nanomaterials on the solid-solution interface within the synergy, enabling a recorded exponential enhancement in the synthesis efficiency and programmable customization of the products' amount, size, and morphology. This work presents a new methodology and technology in both Joule-heating-based and wet-chemical synthesis. Below are the detailed comparison and summary:

1) Recent advances were achieved by Joule-heating-based high and controllable temperature in solid-solid systems

Joule heating represents one of the most cutting-edge and advanced technologies nowadays. Benefiting from its superiority in highly-efficient and precise heat output, it has recently initiated a new era of material synthesis. Generally, these reaction systems are constructed by compressing dry and conductive solid precursors between two electrodes or loading dry solid precursors on the electrothermal substrates. Following an instantaneous Joule heating process, the precursors were thermally decomposed/melted/dispersed, and then assembled/reacted into nanomaterials. It shows

that these advances were achieved by solid-state reactions in the solid-solid systems, but not in the solid-liquid (solution) system for liquid-phase wet-chemical synthesis. In these systems, within the rapid and highly-efficient heat conduction from the electrothermal materials to the solid precursors, it can achieve high temperature, fast heating/cooling, and precise temperature control over the reaction, thereby using Joule heating (temperature effect) to trigger, accelerate and control the synthesis.

2) Obvious differences of WIJH synthesis in the solid-liquid system in terms of thermal conduction, process, and mechanism for crystallization of nanomaterials

However, wet-chemical synthesis, one of the most classic methods to prepare almost all kinds of nanomaterials, has rarely benefited from Joule heating, mainly due to the weakened heat effect in bulk solution. The conditions, processes, and mechanisms for forming nanomaterials from precursor solution are significantly different from those from solid precursors.

Thermal conduction is totally different and more complicated in the solid-liquid (solution) system. When heating liquid (bulk solution), it is hard to achieve high temperature, high ramping/cooling rate, and control of the thermal field. This becomes one of the greatest hindrances to improving the synthesis efficiency and controllability of current wet-chemical synthesis. Through constructing a thin-layer liquid film to confine the Joule heat at the solid-solution interface, our WIJH approach ends the reaction with high temperature (e.g., up to 300 °C for HKUST-1), high ramping (around 300 °C s⁻¹) and cooling rates (above 200 °C s⁻¹), and sensitive modulation/switch of the temperature. Therefore, we realized the trigger, acceleration, stop, and even control of the reaction in solution through Joule heating.

Moreover, besides Joule heating, the accompanying solvent evaporation also plays a critical role in the WIJH synthesis. It could accelerate the ramping, spatially confine the products around the substrate, and, most importantly, the concentration of the precursors. It is absent or ignored in previous works. How the concentration effect is coupled with the high temperature to accelerate and control the synthesis is significant but has rarely been reported before.

With the synergy between Joule heating (temperature effect) and evaporation (concentration effect) in the new confined interfacial heating mechanism, WIJH realized the acceleration and control of the nucleation and growth of the nanomaterials on the solid-solution interface, enabling a series of record (sub-)second wet-chemical syntheses of various materials. The synthesis efficiency is exponentially enhanced when compared with conventional methods. Moreover, via programming the electrified procedure to control the crystallization, WIJH enabled precise control of the products' amount, size, and morphology. To our knowledge, this has never been realized in conventional bulk-heating-based methods, particularly in such a short second-scale time.

In summary, this work is quite different from the previous works on ultrafast synthesis based on Joule heating. We believe **the science of confined wet-interfacial Joule heating to accelerate and control the nucleation and growth, and the combination of super-efficiency, programmability, versatility, and scalability, making our approach a milestone in the research of wet-chemical synthesis and Joule heating,** which is worth publishing in *Nature Communication*.

Table R1 Comparison of the WIJH approach with recent advances in the Joule-heating-based ultrafast synthesis of nanomaterials

Strategy	Reaction system	Mechanism/Process	Material^{reference}
WIJH	Precursor solution on graphene film	The synergy between Joule-heating-based high temperature and evaporation-caused concentration of precursors accelerates and controls the nucleation and growth of the nanomaterials on the solid-solution interface	MOFs, COF, metal, metal oxide, and sulfide (this work)
Carbothermal shock synthesis	Solid metal salts on carbon fiber	Precursors were decomposed into metals under Joule heating, and clustered into nanoparticles during the fast cooling	High-entropy alloy ^{1,2}
Ultrafast high-temperature sintering	Precursor pellets between two carbon papers	Precursors underwent rapid solid-state reaction and reactive sintering under Joule heating	Ceramics ^{3,4}
Flash Joule heating	Carbon sources compressed within two loosely fitting electrodes connecting to a capacitor bank	Carbon sources were heated by high-voltage electric discharge, converting the amorphous carbon into turbostratic flash graphene	Flash graphene ⁵
	Metal precursors and carbon black compressed between two graphite electrodes	Metal precursors were vaporized, and reacted with the solid carbon to form the metal carbides under Joule heating	Transition metal carbides and covalent carbides ⁶
High-temperature shockwave	Solid metal salts on defective carbon	Precursors were decomposed into clusters and single atoms during the on-off Joule heating cycles	Pt, Ru, and Co single atoms ⁷
Ultrahigh-temperature melt printing	Multi-elemental metal powders in a high-temperature column of carbon felt	Multi-elemental metals were melted and mixed into homogenous alloys under Joule heating	Multi-principal element alloys ⁸
Pulsed direct current Joule heating	Carbon black and γ -Al ₂ O ₃ nanoparticles compressed between two graphite electrodes	Rapid phase transformation from γ -Al ₂ O ₃ to α -Al ₂ O ₃ accompanied by the intermediate phase of δ' -Al ₂ O ₃ under the resistive hotspot-induced local heating	α -Al ₂ O ₃ nanoparticles ⁹
Cellulose-derived carbothermal shock	CuCl ₂ -coated cellulose on carbon paper	CuCl ₂ was reduced to Cu, and cellulose was partially carbonized to form carbon nanoclusters for the high-density surface loading of Cu nanoparticles	Single Cu nanoparticles and alloy nanoparticles ¹⁰
Coordinated carbothermal shock	Metal-ligand precursors on carbon paper	Precursors underwent assembly into an ordered structure at low temperature, and was carbonized at high temperature to form nanoparticles on 2D porous carbon	High-density and ultrasmall metal nanoparticles on 2D porous carbon ¹¹

Joule heating	Microparticles on reduced graphene oxide film	Microparticles were melted under high temperature, separated, and self-assembled into nanomaterials on cooling	Metal or silicon nanoparticles ^{12,13}
Low-pressure carbothermal shock	Oxides@carbon pellets between two carbon cloths	Carbothermal reduction of oxide and carbon mixtures under Joule heating in a low-pressure environment	Refractory metal carbides and high-entropy carbides ¹⁴
Thermal shock synthesis/Rapid thermal pulse	Metal precursors on carbon fiber or 3D-printed graphene oxide framework	Precursors were decomposed into metals under Joule heating, and clustered into nanoparticles during the fast cooling	(Multi-)metallic nanoparticles ^{15,16} ; non-noble metal high-entropy oxides ¹⁷ ; Ru nanoparticles ^{18,19} ; Pt nanoparticles ²⁰ ; Ni nanoparticles ²¹
Joule heating	Precursors adsorbed on carbon black between carbon clothes	Ni and N of precursors were evaporated and interacted to form Ni-N _x radicals on the carbon blank, while metal-free N species were decomposed and precluded from carbon black under high temperature	Ni-N _x single-atom ²²
Joule heating	Mixed metal salts on carbon nanofiber	Precursors were decomposed and underwent an atomic ordering transformation (disorder-to-order transition) to form intermetallic nanoparticles under Joule heating	Multi-principal elemental intermetallic nanoparticles ²³ ; intermetallic nanoparticles ²⁴
Flash Joule heating	Catalyst-loaded polymer (plastic) and amorphous carbon black compressed within graphite electrodes	Graphitic 1D nanomaterials were grown through the deposition of mobile carbon on metallic nanoparticles under Joule heating	Flash graphitic 1D material and its hybrid graphitic 1D/2D materials ²⁵
	BH ₃ NH ₃ and carbon black compressed inside a quartz tube between two copper electrodes	BH ₃ NH ₃ was thermally decomposed and dehydrogenated to form BN-based structures under Joule heating	Turbostratic BCN and BN ²⁶
	Carbon sources compressed within two loosely fitting electrodes	Carbon sources were heated by high-voltage electric discharge, converting the amorphous carbon into turbostratic flash graphene	Flash graphene ²⁷⁻³¹
	Carbon black and dopants or single-walled carbon nanotubes compressed within two graphite electrodes	Dopants were volatilized and reacted with the amorphous carbon sources under transient ultrahigh temperature by Joule heating	Heteroatom-doped flash graphene ³² ; fluorinated carbon allotropes ³³ ; flash graphene crosslinked nanoribbons ³⁴

	Carbon black and MoS ₂ or WS ₂ powders compressed with two electrodes	The bulk conversion from 2H phases to 1T phases under Joule heating	Metastable dichalcogenides ³⁵	metal
High-temperature pulse	Metal oxide particles on carbon nanofiber	Large aggregated nanoparticles were redispersed into nanoscale components under short high-temperature pulse	Metallic nanoparticles ³⁶	
Programmable electrothermal waves	Needlelike nanoarrays of Co(OH) ₂ on nickel foam	Co(OH) ₂ was decomposed to cobalt oxides under Joule-heating-driven electrothermal waves	CoO/Co ₃ O ₄ @Ni foam ³⁷	
	Pd(NO ₃) ₂ on carbon fiber	Nitrate was vaporized to NO _x , creating an O-rich environment, and the activated Pd metallic ions were partially oxidized to Pd/Pd _x O _y composites	Heterogeneous Pd/Pd _x O _y /carbon ³⁸	
Joule heating	Manganese and silver nitrates on carbon fiber	Metal nitrates were fully decomposed, and the metal ions were reformed as manganese oxides and silver	Ag-Mn _x O _y branches ³⁹	
	Hydroxide precursors in the folded ship-type nickel foil	Precursors were thermally decomposed by Joule heating	Doped metal oxides ⁴⁰	
Joule heating	Si nanospheres on carbon nanofiber	Si melted and reacted with the carbon nanofiber under high temperature	SiC/C nanocomposites ⁴¹	
Ultrafast Joule heating synthesis	Co ²⁺ -containing graphene oxide aerogel	Joule heating provided the activation energy for the dispersion of metal atoms, the reduction, and doping of graphene oxide	Co-N-C single-atom on graphene ⁴²	porous
High-temperature rapid heating	Mixed metal salts on a folded carbon paper	Precursors were decomposed and formed oxide phase under high-temperature Joule heating	Multi-elemental metal oxide microparticles ⁴³	oxide
High-temperature-pulse	CoCl ₂ on carbon nanofiber or carbonized wood	CoCl ₂ was decomposed and formed Co clusters under Joule heating, and carbon atoms were detached locally from the substrate simultaneously	Ultrathin-graphene-coated nanoparticles ⁴⁴	Co
Thermal Shock	Cobalt-thiourea coordination complex on nickel foam	Ni was activated, and the complex was converted to doped carbon-coated CoS by Joule heating	Doped carbon-coated CoS ⁴⁵	
High-temperature shock carbonization	Precarbonized coconut shells and KOH powders on the carbon cloth	Coconut shells were precarbonized, activated by KOH, and finally formed activated porous carbons under thermal shock	Activated porous carbons ⁴⁶	
Joule heating/High-temperature pulse	Microparticles on reduced graphene oxide film	Microparticles were melted under high temperature, separated, and self-assembled into nanoparticles on cooling	Silicon, tin and aluminum ⁴⁷ ; FeS ₂ ⁴⁸ , and Co ₂ B, MoS ₂ , Co ₃ O ₄ ⁴⁹	
“Droplet-to-particle”	Precursors fly through the carbonized	Precursors were decomposed under high temperature, and formed nanoparticles	Multielement high entropy	entropy

spray pyrolysis	wood micro-channel reactor		alloy/oxides ⁵⁰
Pulsed physical vapor deposition	Tellurium powders on the pulse Joule-heated mica	Joule heating induced thermal evaporation of tellurium thin film to form nanowires on the above accepting substrate	Ultrathin Te nanowires ⁵¹

References

- [1] Yao, Y. et al. Carbothermal shock synthesis of high-entropy-alloy nanoparticles. *Science* **359**, 1489-1494 (2018).
- [2] Xie, P. et al. Highly efficient decomposition of ammonia using high-entropy alloy catalysts. *Nat. Commun.* **10**, 4011 (2019).
- [3] Wang, C. et al. A general method to synthesize and sinter bulk ceramics in seconds. *Science* **368**, 521-526 (2020).
- [4] Wang, R. et al. Computation-guided synthesis of new garnet-type solid-state electrolytes via an ultrafast sintering technique. *Adv. Mater.* **32**, e2005059 (2020).
- [5] Luong, D. X. et al. Gram-scale bottom-up flash graphene synthesis. *Nature* **577**, 647-651 (2020).
- [6] Deng, B. et al. Phase controlled synthesis of transition metal carbide nanocrystals by ultrafast flash Joule heating. *Nat. Commun.* **13**, 262 (2022).
- [7] Yao, Y. et al. High temperature shockwave stabilized single atoms. *Nat. Nanotechnol.* **14**, 851-857 (2019).
- [8] Wang, X. et al. Ultrahigh-temperature melt printing of multi-principal element alloys. *Nat. Commun.* **13**, 6724 (2022).
- [9] Deng, B. et al. High-surface-area corundum nanoparticles by resistive hotspot-induced phase transformation. *Nat. Commun.* **13**, 5027 (2022).
- [10] Song, J. Y. et al. Generation of high-density nanoparticles in the carbothermal shock method. *Sci. Adv.* **7**, eabk2984 (2021).
- [11] Shi, W. et al. Transient and general synthesis of high-density and ultrasmall nanoparticles on two-dimensional porous carbon via coordinated carbothermal shock. *Nat. Commun.* **14**, 2294 (2023).
- [12] Chen, Y. et al. Ultra-fast self-assembly and stabilization of reactive nanoparticles in reduced graphene oxide films. *Nat. Commun.* **7**, 12332 (2016).
- [13] Li, Y. et al. In Situ, fast, high-temperature synthesis of nickel nanoparticles in reduced graphene oxide matrix. *Adv. Energy Mater.* **7**, 1601783 (2017).
- [14] Han, Y. C. et al. A general method for rapid synthesis of refractory carbides by low-pressure carbothermal shock reduction. *Proc. Natl. Acad. Sci. U.S.A.* **119**, e2121848119 (2022).
- [15] Huang, Z. et al. Direct observation of the formation and stabilization of metallic nanoparticles on carbon supports. *Nat. Commun.* **11**, 6373 (2020).
- [16] Lacey, S. D. et al. Stable multimetallic nanoparticles for oxygen electrocatalysis. *Nano Lett.* **19**, 5149-5158 (2019).
- [17] Abdelhafiz, A. et al. Carbothermal shock synthesis of high entropy oxide catalysts: dynamic structural and chemical reconstruction boosting the catalytic activity and stability toward oxygen evolution reaction. *Adv. Energy Mater.* **12**, 2200742 (2022).
- [18] Qiao, Y. et al. Transient, in situ synthesis of ultrafine ruthenium nanoparticles for a high-rate Li-CO₂ battery. *Energy Environ. Sci.* **12**, 1100-1107 (2019).
- [19] Qiao, Y. et al. Thermal shock synthesis of nanocatalyst by 3D-printed miniaturized reactors. *Small* **16**, e2000509 (2020).

- [20] Zhou, Y. et al. Tuning the high-temperature wetting behavior of metals toward ultrafine nanoparticles. *Angew. Chem. Int. Ed.* **57**, 2625-2629 (2018).
- [21] Qiao, Y. et al. 3D-printed graphene oxide framework with thermal shock synthesized nanoparticles for Li-CO₂ batteries. *Adv. Funct. Mater.* **28**, 1805899 (2018).
- [22] Xi, D. et al. Limiting the uncoordinated N species in M-N_x single-atom catalysts toward electrocatalytic CO₂ reduction in broad voltage range. *Adv. Mater.* **34**, e2104090 (2022).
- [23] Cui, M. et al. Multi-principal elemental intermetallic nanoparticles synthesized via a disorder-to-order transition. *Sci. Adv.* **8**, eabm4322 (2022).
- [24] Cui, M. et al. Rapid atomic ordering transformation toward intermetallic nanoparticles. *Nano Lett.* **22**, 255-262 (2022).
- [25] Wyss, K. M. et al. Upcycling of waste plastic into hybrid carbon nanomaterials. *Adv. Mater.* e2209621 (2023).
- [26] Chen, W. et al. Turbostratic boron-carbon-nitrogen and boron nitride by flash Joule heating. *Adv. Mater.* **34**, e2202666 (2022).
- [27] Algozeeb, W. A. et al. Flash graphene from plastic waste. *ACS Nano* **14**, 15595-15604 (2020).
- [28] Stanford, M. G. et al. Flash graphene morphologies. *ACS Nano* **14**, 13691-13699 (2020).
- [29] Saadi, M. et al. Sustainable valorization of asphaltene via flash Joule heating. *Sci. Adv.* **8**, eadd3555 (2022).
- [30] Beckham, J. L. et al. Machine learning guided synthesis of flash graphene. *Adv. Mater.* **34**, e2106506 (2022).
- [31] Huang, P. Effect of free radicals and electric field on preparation of coal pitch-derived graphene using flash Joule heating. *Chem. Eng. J.* **450** (2022).
- [32] Chen, W. et al. Heteroatom-doped flash graphene. *ACS Nano* **16**, 6646-6656 (2022).
- [33] Chen, W. et al. Ultrafast and controllable phase evolution by flash Joule heating. *ACS Nano* **15**, 11158-11167 (2021).
- [34] Advincula, P. A. et al. Tunable hybridized morphologies obtained through flash Joule heating of carbon nanotubes. *ACS Nano* **17**, 2506-2516 (2023).
- [35] Chen, W. et al. Millisecond conversion of metastable 2D materials by flash Joule heating. *ACS Nano* **15**, 1282-1290 (2021).
- [36] Xie, H. et al. High-temperature pulse method for nanoparticle redispersion. *J. Am. Chem. Soc.* **142**, 17364-17371 (2020).
- [37] Kim, W. et al. Precisely tunable synthesis of binder-free cobalt oxide-based Li-ion battery anode using scalable electrothermal waves. *ACS Nano* **16**, 17313-17325 (2022).
- [38] Seo, B. et al. Electrothermally tunable morphological and redox design of heterogeneous Pd/Pd_xO_y/carbon for humidity-hydrion-driven energy harvesters. *Nano Energy* **95**, 107053 (2022).
- [39] Yeo, T. et al. Ultrafast extreme thermal-electrical fabrication of volcano-shape-like core-shell Ag-Mn_xO_y branches anchored on carbon as high-performance electrochemical electrodes. *Nano Energy* **91**, 106663 (2022).
- [40] Li, Y. et al. Rapid synthesis of doped metal oxides via Joule heating for oxygen electrocatalysis regulation. *J. Mater. Chem. A* **11**, 10267-10276 (2023).
- [41] Xie, H. et al. Necklace-like silicon carbide and carbon nanocomposites formed by steady Joule heating. *Small Methods* **2**, 1700371 (2018).

- [42] Xing, L. et al. Ultrafast Joule heating synthesis of hierarchically porous graphene-based Co-N-C single-atom monoliths. *Nano Res.* **15**, 3913-3919 (2021).
- [43] Dong, Q. et al. Rapid synthesis of high-entropy oxide microparticles. *Small* **18**, e2104761 (2022).
- [44] Xie, H. et al. High-temperature-pulse synthesis of ultrathin-graphene-coated metal nanoparticles. *Nano Energy* **80**, 105536 (2021).
- [45] Wu, H. et al. Thermal shock-activated spontaneous growing of nanosheets for overall water splitting. *Nano-micro Lett.* **12**, 162 (2020).
- [46] Liu, Z. et al. Ultrafast porous carbon activation promises high-energy density supercapacitors. *Small* **18**, e2200954 (2022).
- [47] Chen, Y. et al. Rapid, in situ synthesis of high capacity battery anodes through high temperature radiation-based thermal shock. *Nano Lett.* **16**, 5553-5558 (2016).
- [48] Chen, Y. et al. FeS₂ nanoparticles embedded in reduced graphene oxide toward robust, high-performance electrocatalysts. *Adv. Energy Mater.* **7**, 1700482 (2017).
- [49] Xu, S. et al. Universal, in situ transformation of bulky compounds into nanoscale catalysts by high-temperature pulse. *Nano Lett.* **17**, 5817-5822 (2017).
- [50] Wang, X. et al. Continuous 2000 K droplet-to-particle synthesis. *Mater. Today* **35**, 106-114 (2020).
- [51] Li, S. et al. Programmable nucleation and growth of ultrathin tellurium nanowires via a pulsed physical vapor deposition design. *Adv. Funct. Mater.* 2211527 (2022).

1. Considering the previous publications on the Joule heating method, the authors are required to add more recent related progresses. Moreover, the main differences and the novelty of this work should be highlighted in the introduction part.

R: Thank you. We revised the introduction according to the reviewer's comments and the above discussion. In the revised manuscript, **the recent significant advances** in Joule-heating-based synthesis have been introduced and cited. **The differences and the novelty of our WIJH approach** have been emphasized **in terms of the unique reaction system of the solid-solution interface, and the new mechanism of confined interfacial heating of the synergy of temperature and concentration effects**. These are different from recent advances achieved by the solid-solid systems, in which high and controllable temperature accounts for the acceleration and control of the synthesis. Besides, **the results of the exponential enhancement in the synthesis efficiency and the programmability** towards the products' amount, size, and morphology have also been highlighted.

2. Fig. 2c and 2b are hard to understand and authors should add the relevant description of the symbols in the Figures, such as the C , C_s , and h_0 . In addition, the detailed experiment content about how to get the h in Fig. 2b and how to get the density in Fig. 2c should be provided in manuscript or supplementary information.

R: Thank you. Figs. 2b and 2c have been updated. The corresponding description of the symbols and the details to obtain them have also been added in the revised manuscript and the supplementary information.

In Fig. 2b (i.e., Fig. R1-a), the blue part displays the schematic illustration of the LaMer model, which is a typical diagram in the synthesis of nanoparticles to show the relationship between nucleation/growth and concentration of monomers (reactants) (*Angew. Chem. Int. Ed.* **60**, 26390 (2021); *Adv. Mater.* **30**, e1800202 (2018)). c_n and c_s represent the critical concentration of nucleation and the saturation concentration (at which the growth rate equilibrates with the solvation rate), respectively. The red part is the normalized height of the liquid film during the WIJH process, which was used to indicate the concentration change to analyze the concentration effect. h_0 in the original manuscript is the initial height of the liquid film, which was measured to be $\sim 165 \mu\text{m}$ by comparing the photographs before and after the spreading of the liquid on the GF (Fig. R2, using a Video contact Angle analyzer (Dataphysics, OCA20)). The normalized height was obtained by the simulation of the evaporation in the WIJH, as it is hard to precisely record by experiment within a low height changing from several micrometers to nanometers in a short sub-second duration.

Particle density in Fig. 2c (Fig. R1-b) was calculated as the statistical quantity of the HKUST-1 particles divided by the area of the GF according to the SEM images of the HKUST-1/GF sample (Fig. R3). The particle densities were calculated to be 331 ± 165 , 1636 ± 149 , 2199 ± 134 , 1399 ± 101 , 667 ± 50 , and 600 ± 28 particles per μm^2 GF at 0.25, 0.4, 0.55, 0.7, 0.85, and 0.95 s, respectively. To clearly display the trend of the nucleation in the diagram, these particle densities were further normalized to that of the sample prepared at the initial stage (0.25 s). It is a common method in other works (e.g., *Sci. Adv.* **6**, eabd4045 (2020)).

Fig. R1 a) Schematic of the LaMer model (blue region, where c_n and c_s represent the critical and the saturation concentration of nucleation, respectively), and the normalized height of the liquid film (red region, which was obtained by the simulation). b) A conclusive formation diagram of the nucleation and growth, according to the statistics of the particle densities and sizes from the SEM images. The particle densities were normalized to the value of the sample prepared within 0.25 s.

Fig. R2 Optical images of the cross-section of GF a) before and b) after the addition of 2 μL of the mixture of H_2O , EtOH , and DMF (1:1:1). The height of the liquid film was measured by comparing the photographs before and after the spreading of the liquid on the GF.

Fig. R3 SEM images of HKUST-1/GF obtained by WIJH with a) 0.25 s, b) 0.4 s, c) 0.55 s, d) 0.7 s, e) 0.85 s, f) 0.95 s. Scale bar: 2 μm .

3. Authors have discussed the WIJH synthesis mechanism and the difference on the growth of HKUST-1 for the various synthesis conditions, have some different found in nucleation or growth mechanism for the WIJH compared with the conventional bulk-heating-based methods or solvothermal method?

R: Thank you. Temperature and concentration are essential elements of reaction kinetics and thermodynamics that affect the crystallization. The WIJH approach proposed a new confined interfacial heating mechanism to accelerate and control the nucleation and growth of the nanomaterials on the solid-solution interface. **There are three obvious differences in nucleation and growth mechanism/process**, when compared with conventional methods: 1) **the synergistic mechanism between enhanced temperature and concentration**; 2) **the successive, ultrafast, and programmable nucleation and growth processes**; 3) **the formation for the final product of a flat and fusing film of one-layer HKUST-1 particles**. In conventional bulk-heating-based methods, within a slow heating process and drastically-decreased concentration of the precursors, nucleation, and growth are simultaneous, slow, and random, and the final products present as rugged films of randomly stacked or intergrown particles. Below are the detailed comparison and discussion.

In conventional bulk-heating-based methods, including solvothermal synthesis, **heating is a slow and near-equilibrium process** with the thermal conduction from the bulk solution to the specific substrate. Therefore, the thermal field for the reaction is determined by the heat-related properties of the bulk solution, which generally presents as limited temperature (<200 °C, the boiling point of the solvent), low ramping (e.g., 4 °C min⁻¹ for a solvothermal autoclave) and cooling rates, as well as the poor controllability of the thermal distribution. As for **the concentration of the reactants, it drops rapidly and dramatically as the reaction consumes in the sealed reaction system**. The limited heating and concentration would slow down the crystallization kinetics considerably (*Chem. Rev.* **112**, 933, (2012); *J. Mater. Chem. A* **8**, 7633 (2020)). Particularly for the synthesis on the substrate with limited nucleation sites, the heterogeneous crystallization on the substrate was further hindered by the preferential and competitive consumption of energy and precursors by the homogeneous crystallization in the bulk solution. Therefore, these methods generally suffer from long reaction time (several hours to days), poor controllability, and heavy consumption of energy and reactants. Besides, **the nucleation and growth occur simultaneously and randomly**, producing the products of rugged films of randomly stacked or intergrown particles with a broad size distribution.

In WIJH synthesis, stemming from the powerful electrothermal effect of the GF and a thin-layer liquid film for heat confinement, the WIJH system possesses high reaction temperature (e.g., up to 300 °C for the synthesis of HKUST-1), high heating rate (around 300 °C s⁻¹) and cooling rate (approximately 200 °C s⁻¹). The simultaneous evaporation of the solvent can concentrate the precursors, which timely compensates the consumed precursors to ensure a relatively-high concentration. **With the synergy between the enhanced temperature and concentration, the nucleation and growth events are successive, and exponentially accelerated from several hours to sub-seconds**. The high temperature by Joule heating of the GF induces the initial nucleation, The drastic evaporation caused by Joule heating markedly elevates the concentration of the precursors, and thus, induces the burst of the nucleation. In the following growth stage, with the synergy of the concentrated precursors by ongoing evaporation, high temperature at the plane interface enables fast growth and induces the formation of the unique fusing film. **This synergistic mechanism is new and apparently different from conventional bulk heating.**

Besides, as the heat was generated and confined around the substrate interface, the nucleation and growth on the substrate were significantly promoted and controlled by the modulation of the temperature (programming the electrified procedures). As a result, WIJH enabled the customization of the products' amount, size, and morphology. To our knowledge, **this has never been realized in conventional bulk-heating-based methods, particularly in a short second-scale time.**

In summary, WIJH realized the acceleration and control of the nucleation and growth on the solid-solution interface based on a unique mechanism of confined interfacial heating of the synergy between temperature and concentration effects. It endows the WIJH approach with super-efficiency, controllability, and universality for the wet-chemical syntheses of various nanomaterials. The as-prepared HKUST-1/GF film consisting of one-layer fusing particles displayed superior performances and efficiency in the controllable capture and liberation towards CO₂. The above discussion has been summarized and emphasized in the revised manuscript.

4. How did authors obtain the specific value of the normalized quantity in Fig. 3a?

R: Thank you. The quantity (revised as density) in Fig. 3a (i.e., Fig. R4) was obtained as the statistical quantities of the HKUST-1 particles divided by the area of the GF, according to the SEM images of different HKUST-1/GF samples obtained with one to six pulses in Fig. R5. The particle densities were calculated to be 0 , 47 ± 21 , 239 ± 100 , 808 ± 197 , 1847 ± 250 , and 2988 ± 510 particles per μm^2 GF by 1, 2, 3, 4, 5, and 6 pulses, respectively. Afterward, these densities were normalized to that of the sample prepared by two pulses.

We revised “the normalized quantity” as “the normalized particle density” to avoid ambiguity. The detailed calculation method was added in the characterization part of the supplementary information.

Fig. R4 Pulse-cycle-based modulation towards particle density. From top to bottom are the electrified procedure (the current pattern), the temperature profile of the WIJH system, statistical results of the particle density that normalized to the value of the sample prepared within two pulses, and the typical SEM images of the HKUST-1/GF (the border color of the image was used to mark the corresponding sample in the statistical result, scale bar: 1 μm for all).

Fig. R5 SEM images of HKUST-1/GF obtained with a) 1, b) 2, c) 3, d) 4, e) 5, and f) 6 pulses.

5. Achieving the large-scaled synthesis using Joule heating method is significant for industrial production, thus I recommend that the authors provide more experiment content and discussion on the roll-to-roll Joule-heating continuous fabrication.

R: Thank you. We have supplemented the detailed experimental parameters and additional experiments on the roll-to-roll continuous fabrication in the revision. The practicability of the continuous fabrication of a fusing film, dispersed nanoparticles, and a multi-layer intergrown film has been investigated. Besides, we have developed a layer-by-layer fashion to further control and scale up the fabrication for potential industrial production. Below are the details:

Experiment content:

The continuous production was demonstrated on a roll-to-roll Joule-heating fabrication system. The system mainly consists of two pairs of parallel graphite electrodes (interval distance of 7 cm), two controllable micromotors to roll the electrodes with the speed of 10 rpm, and a d.c. power source (ITECH, IT65220) connecting with the electrodes via two electric brushes (Fig. R6-a). A continuous GF strip with a width of 2 cm was clamped and passed through the rotating electrodes. The precursor solution of HKUST-1 (85 mM $\text{Cu}(\text{NO}_3)_2$ and 55 mM H_3BTC) was spread on the GF to form a thin-layer liquid film (around $10 \mu\text{L cm}^{-2}$, on the area between the electrodes). Afterward, an instantaneous current flow (24 A, 6 V) was applied to the electrodes to conduct the WIJH synthesis. Finally, the film was collected, followed by washing and drying. For the layer-by-layer fashion, the HKUST-1-coated GF was cycled into the area between two electrodes by the micromotors. A series of repeating procedures of the addition of the precursor solution on the same region of the GF and the WIJH synthesis were conducted successively until completing the fabrication.

Results and discussion:

With the aid of a roll-to-roll Joule-heating setup, the WIJH approach can be scaled up readily for the continuous and mass fabrication of nanomaterials on the substrate, providing potential opportunities for industrial-scale production. For one batch of fabrication of HKUST-1/GF, both fusing HKUST-1 film and dispersed nanoparticles could be obtained (Figs. R6-b and R6-c), giving the production rates around $14.7 \text{ cm}^2 \text{ s}^{-1}$ and $25 \text{ cm}^2 \text{ s}^{-1}$, respectively. MOF loading can be further scaled up and easily adjusted through a repeating layer-by-layer fashion, producing intergrown multi-layer HKUST-1 coatings on the GF (Fig. R6-d). From the first to five cycles, the loading mass rises from (1.13 ± 0.33) to $(4.67 \pm 0.38) \text{ g m}^{-2}$ (Fig. R6-e).

Fig. R6 a) Photographs of the setup for the roll-to-roll continuous fabrication for HKUST-1/GF. SEM images of a) the obtained fusing film and b) dispersed nanoparticles. c) SEM images of the multi-layer intergrown film obtained by the repeating layer-by-layer fashion. Inset is the cross-sectional SEM image. d) Areal loading of HKUST-1 on the GF in the layer-by-layer continuous fabrication.

6. As the control experiment, the performance of the CO₂ capture and liberation for the HKUST-1/GF synthesized by conventional method should be added and discussed.

R: Thank you. To evaluate the performance, we have synthesized HKUST-1/GF by the typical solvothermal method (marked as HKUST-1/GF_{solvothermal}, 85 mM Cu(NO₃)₂ and 55 mM H₃BTC, at 120 °C for 12 h), and collected its CO₂ adsorption isotherm without (capture) and with Joule heating (liberation). Due to the limited loading of HKUST-1 by the solvothermal method, HKUST-1/GF_{solvothermal} presented a lower adsorption capacity of 5.73 cm³ g⁻¹ at 298 K and 1 bar than that of HKUST-1/GF_{WIJH} (14.56 cm³ g⁻¹). Similar to the HKUST-1/GF_{WIJH}, the IJH-controlled liberation of CO₂ has been successfully demonstrated. Below is the detail:

As shown in Fig. R7, the capture capacities of GF, and HKUST-1/GF that prepared by solvothermal and the WIJH methods were 2.8, 5.73, and 14.56 cm³ g⁻¹ at 298 K and 1 bar, respectively. The capacity of HKUST-1/GF_{WIJH} is much higher than that of HKUST-1/GF_{solvothermal},

which is ascribed to the relatively high mass and areal loading of HKUST-1 film on the GF by WIJH synthesis. In contrast, a few and nonuniform loading of HKUST-1 was found on the GF obtained by the solvothermal method, in which more HKUST-1 formed in the bulk solution (Fig. R8). The capacities of HKUST-1_{WIJH} and HKUST-1_{solvothermal} were correspondingly calculated to be 3.95 and 3.78 mmol g⁻¹, which are comparable with the values reported by other works (*Chem. Eng. J.* **420**, 129677 (2021); *Chem. Eng. J.* **413**, 127396 (2021)). In addition, as shown in Fig. R7-b, the highly-efficient and programmable liberation of the adsorbed CO₂ on the HKUST-1/GF_{solvothermal} was also successfully realized, indicating the universality of IJH strategy for the capture and liberation of gas. The above results and discussion have been supplemented in the revised manuscript and supplementary information.

Fig. R7 a) CO₂ adsorption isotherms of GF, HKUST-1/GF prepared by WIJH and solvothermal synthesis at 25 °C. b) CO₂ adsorption isotherm of HKUST-1/GF_{solvothermal} under programmed IJH processes.

Fig. R8 Photograph of the HKUST-1/GF synthesized by the solvothermal method. The white arrows indicate the few and nonuniform loading of HKUST-1 (blue) on the GF (gray).

Responses to Reviewer #2:

This manuscript reports the synthesis of materials using Joule heating method. These materials are often prepared by the wet-chemistry that involves a longer synthesis time and large usage of reactants/solvent etc. By placing a thin layer of synthesis precursor on the heating medium, a carbon file, the authors claimed that the materials including MOFs, COF, metal, metal oxide and sulfide were prepared within sub-second. Overall this is a very interesting work since the synthesis method is new and the formed materials morphology is quite different from the conventional thermal synthesis.

We sincerely appreciate the reviewer for the positive and valuable comments on the manuscript. We have tried our best to address these issues. A point-by-point response to the reviewer's comments is provided below.

Specific comments are listed below.

(1) The materials synthesized by this Joule heating method are simple and easy to crystallize. For example, ZIF-8, HKUST etc does not need very harsh conditions to synthesize. From this point of view, it is hard to see the obvious advantages of the Joule heating method for wet-chemistry synthesis of solid materials.

R: Thank you. The reason why we chose HKUST-1 as the model to display the WIJH synthesis is that it is one of the most extensively studied MOFs materials. Moreover, it is not very easy to be synthesized on the specific substrate, which generally requires a solvothermal reaction around/above 100 °C for up to 56 h. **Compared with current methods for synthesizing HKUST-1 on the substrate, WIJH presents apparent advantages in the superefficient synthesis of exponentially reduced time, energy and reactants costs, and programmability towards the nucleation and growth.** Beyond HKUST-1, **other materials that generally require harsh conditions for crystallization**, for instance, heating above 120 °C for 3-7 days to form TPB-DMTP-COF, and heating at 80-120 °C for up to 24 h to form MIL-88A, **have also been successfully synthesized in a sharply-decreased second-scale time.** These also illustrate the advantages and the universality of the WIJH method to synthesize various nanomaterials under harsh or mild conditions. Below is a detailed comparison.

1) Exponentially improved synthesis efficiency of HKUST-1 on the substrate

Following the reviewer' comment, we have taken a comprehensive survey on the crystallization conditions for HKUST-1 on the substrate (Supplementary Table 2). Generally, heating and/or surface modification/activation of the substrate have been widely adopted, as it is not very easy to form HKUST-1 on the substrate, especially for inert substrates like graphene film with limited nucleation sites (*Adv. Mater.* **27**, 7293 (2015); *J. Mater. Chem. A* **5**, 1948 (2017); *ACS Appl. Mater. Interfaces* **11**, 22714 (2019)). The synthesis begins with copper salt and H₃BTC in water, ethanol, and DMF solvents, and is conducted around/above 100 °C for up to 56 h. **When comparing with these previously-reported methods, WIJH exhibited remarkable advantages in the ultrafast growth rate of 1.97 μm s⁻¹ (up to 10⁵ times faster), the ultralow energy cost of 9.55 × 10⁻⁶ kWh cm⁻² (down to 10⁻⁶ times lower), and ultrahigh mass production efficiency of 1660 μg s⁻¹ (up to 10⁴ times higher) without any surface treatments** (Fig. 1h and Supplementary Table 2).

Control experiments also confirmed the advantage. As shown in Fig. R9-a, under the typical solvothermal conditions, more blue precipitations of HKUST-1 were found in the solution rather than on the GF, leaving an ultralow surface coverage ratio of HKUST-1 around 6% after 60 min. In

contrast, 100% coverage of HKUST-1 on the GF could be achieved as fast as 0.25 s by WIJH synthesis (Fig. R9-b). The low efficiency of solvothermal synthesis is mainly ascribed to the priority of homogeneous crystallization in heating bulk solution, and the limited reaction temperature and drastically-decreased concentration of the precursors limit the kinetics considerably.

Besides, WIJH enabled the crystallization of HKUST-1 on the GF with ultralow concentrations of the precursors (0.85 mM $\text{Cu}(\text{NO}_3)_2$ and 0.57 mM H_3BTC , Fig. R9-d), which is two hundred times lower than conventional methods. In contrast, via conventional solvothermal synthesis, no particles could be found on the GF or in the solution within the same conditions (Fig. R9-c). The above clearly presents the advantages of the WIJH approach in ultrahigh synthesis efficiency of exponentially reduced time, energy and reactants costs.

Fig. R9 SEM images of HKUST-1/GF obtained by solvothermal synthesis at a) 120 °C for 60 min (85 mM $\text{Cu}(\text{NO}_3)_2$ and 55 mM H_3BTC), and c) 120 °C for 60 min (0.85 mM $\text{Cu}(\text{NO}_3)_2$ and 0.56 mM H_3BTC). SEM images of HKUST-1/GF obtained by WIJH synthesis under b) 5.25 A for 0.25 s (85 mM $\text{Cu}(\text{NO}_3)_2$ and 55 mM H_3BTC), and d) 3 A for 0.95 s (0.85 mM $\text{Cu}(\text{NO}_3)_2$ and 0.56 mM H_3BTC).

2) Innovative programmability of the wet-chemical synthesis in the second-scale time

Beyond the improvement in growth rate, WIJH also realized the control of nucleation and growth on the solid-solution interface via programming the electrified procedures. To our knowledge, this has never been realized in conventional bulk-heating-based methods, particularly in a short second-scale time. This could be used to customize the products' amount, size, and morphology.

3) Besides HKUST-1, other materials that required harsh conditions were also synthesized in (sub-)seconds

Besides, other materials that require heating for several minutes to hours to crystallize, including ZIF-8, MIL-88A, TbBTC, TPB-DMTP-COF, Au, MnO_2 , and CdS, have also been synthesized by the WIJH approach in a sharply-decreased (sub-)second time. Particularly, COF is not easy to crystallize, which generally requires harsh conditions of heating above 120 °C for 3-7 days or powerful energy input to promote the crystallization (*Nat. Chem.* **7**, 905 (2015); *Nat. Syn.* **1**, 87 (2022)). However, the crystallization was realized in 0.95 s by WIJH.

(2) The materials' synthesis solution is quick small. During the Joule heating, the evaporation of solvent should have a significant effects on the materials crystallization and growth. The authors mentioned a bit on this effect, but I think this effect should be more obvious than what has been claimed in the manuscript. Control experiments should be done to clarify the dominating mechanism (e.g., Joule heating versus solvent evaporation) in the synthesis.

R: Thank you. We agree with the reviewer that solvent evaporation significantly affects material crystallization, which is also a feature of our WIJH approach. **Evaporation could assist in the rapid temperature ramping, spatially confine the products toward the substrate to increase the load, and concentrate the precursors to accelerate the crystallization.** Precursor concentration is the most crucial effect of evaporation in the WIJH synthesis, and is obviously different from that of conventional sealed bulk-heating-based reaction systems, where the monomer concentration decreases rapidly and dramatically as the reaction proceeds.

Temperature (heating) and concentration (evaporation) are the two most important factors of the confined interfacial heating mechanism for ultrafast yet programmable crystallization. They **affect and are coupled together along the whole WIJH process, synergistically accelerating and controlling the crystallization.** Specifically, they display different characteristics in different crystallization stages according to our recording along the WIJH process (Fig. 2a and Fig. 2b): a rapidly increased temperature caused by Joule heating in incubation-I, an evaporation-caused reduction of the solvent and the corresponding sharply elevated monomer concentration in nucleation-II, and another dramatically increased temperature in growth-III. Therefore, we have tried our best to conduct a series of control experiments to simulate these characteristics, and their effects have been confirmed respectively. Below is the detailed discussion, using the typical WIJH synthesis of HKUST-1/GF under 3 A as the model:

Fig. R10 SEM images of HKUST-1/GF samples obtained by bulk Joule heating under different temperatures of a) 240 °C, b) 300 °C, c) 380 °C, and d) 450 °C.

Incubation stage I (the initial 0.25 s): within powerful and rapid Joule heating, temperature rapidly increases from 23.9 °C to 115 °C. Monomer concentration also increases after the mixture but is generally lower than the critical concentration of nucleation within limited evaporation (the normalized height of the liquid film decreases to 81%). Therefore, the Joule-heating-caused high temperature is expected to dominate this stage, inducing the initial crystallization. To confirm its

effect, the bulk Joule heating under different temperatures were conducted by applying different current intensities to the immersed GF in the bulk solution (85 mM Cu(NO₃)₂ and 55 mM H₃BTC). In that case, the evaporation-caused concentration could be negligible due to a large-volume solution of 200 μL and a short heating time of 5 s. As shown in Fig. R10, as the final temperature of the GF increased from 240 to 450 °C, more and bigger particles appeared on the GF. This demonstrated the Joule-heating-induced crystallization and highlighted one of the advantages of interfacial heating to control the reaction around the GF.

Nucleation stage II (0.25-0.55 s): According to the following classic nucleation equation, the largest effect on nucleation rate comes from supersaturation (related to concentration) (*Chem. Rev.* **114**, 7610 (2014)).

$$\frac{dN}{dt} = A \exp\left(-\frac{16\pi\gamma^3 v^2}{3k^3 T^3 (\ln S)^2}\right)$$

In our case, temperature remains almost unchanged at around 120 °C, leading to a rapid and intense evaporation event in this stage (the normalized height of the liquid film sharply decreased from 81% to 22%). This is expected to markedly elevate the monomer concentration to exceed the critical nucleation concentration and cause the nucleation burst. Therefore, we have designed control experiments to confirm the concentration effect via comparing the nucleation rates obtained by different initial concentrations of the precursors under the same temperature (Fig. R11). The increasing precursor concentrations were used to simulate the evaporation-caused concentration. As the concentrations of Cu(NO₃)₂ increased from 1.7 to 85 mM, the average nucleation rates increased from 85.5 to 318.2 nm s⁻¹, confirming the concentration effect in the nucleation stage.

Fig. R11 The relationship between nucleation rate and initial concentrations of precursors (with the same 3:2 molar ratio of Cu(NO₃)₂ and H₃BTC).

Growth stage III (0.55-0.95 s): Due to the rapid consumption by nucleation, monomer concentration declines to a level below the critical concentration of nucleation. While the temperature of the WIJH system increased sharply within the ongoing heat input. Therefore, we investigate the temperature effect on the growth, by collecting the products obtained within the same nucleation conditions but at different growth temperatures. These experiments were conducted by programming the electrified procedures of an initial pulse of 3 A for 0.55 s for nucleation, followed by the pulse of different current intensities to achieve different temperatures (Fig. R12-a). As shown in Fig. R12-b, the average growth rates increased from 227.5 to 1697.5 nm s⁻¹, as the final temperature increased from 166 to 289 °C. It confirms that the temperature plays a critical role in the growth. Meanwhile, the ongoing evaporation could further compensate for the consumption of the precursors, enabling a high concentration for fast growth. Besides, as discussed in the original

manuscript, the high-temperature plane of GF also mainly induced the formation of a unique fusing film consisting of one-layer particles.

Fig. R12 a) Temperature profiles of the WIJH synthesis within different growth temperatures. b) The relationship between growth rate and growth temperature. SEM images of the corresponding products of HKUST-1/GF obtained with different growth temperatures of c) 289 °C, d) 242 °C, e) 198 °C, and f) 166 °C.

In summary, evaporation is the result of confined interfacial heating in an open reaction system, which promotes crystallization in terms of rapid ramping, spatial confinement, and precursor concentration. As two essential elements of reaction kinetics, temperature (heating) and concentration (evaporation) accelerate and control the WIJH synthesis synergistically. Control experiments confirmed the Joule heating effect in initial incubation, evaporation-caused concentration effect on the burst of nucleation, and high-temperature-induced growth and formation of the unique fusing film. Note that **although we have discussed the most remarkable effects in different stages respectively, temperature and concentration actually participate and affect the whole WIJH crystallization process. As they are coupled together and affect mutually, it is difficult to clearly distinguish the specific contribution of high temperature and concentration on the crystallization.** The above new experimental results and discussion were supplemented in the revised manuscript and supplementary information.

(3) In line 157, the nucleation equation, I think there should be a negative sign in the exponent. Otherwise, the dependence on Temperature and supersaturation does not seem to be correct.

R: Thank you. Exactly, there should be a negative sign in the nucleation equation. Sorry for the mistake, and we have revised the equation as follows:

$$\frac{dN}{dt} = A \exp\left(-\frac{16\pi r^3 v^2}{3k^3 T^3 (\ln S)^2}\right)$$

(4) How did the authors get the growth rate in the synthesis? The heating and cooling is fast which needs techniques to capture these temperature profiles precisely. On the basis of this, is it possible to get the rate. How does the temperature ramp up and down affect the growth?

R: Thank you. **The rate was calculated as the average size of the HKUST-1 particle divided by the synthesis time.** The size was measured by the sample's SEM image using the ImageJ software. The time was precisely controlled and recorded via the electrified procedure with a DC power source (ITECH, IT6431, temporal resolution of 1 ms). The current output mode was adopted for supplying with basic parameters of time, current, resistance of 0 Ω and the maximum output voltage of 15 V. The given time is regarded as the synthesis time, since the reaction could be stopped timely within a high cooling rate of 200 $^{\circ}\text{C s}^{-1}$ after cutting off the current (Fig. R13). Besides, **the temperature profile and thermal image of the WIJH system were in-situ recorded in a top view by a high-speed infrared thermometer** with a temporal resolution of 67 ms (Teledyne FLIR, T630sc). The above methods are well-developed in previous reports on Joule-heating-based synthesis (e.g., *Nature* **577**, 647 (2020); *Nat. Nanotechnol.* **14**, 851 (2019); *Nat. Commun.* **14**, 2294 (2023)).

Stemming from the unique confined interfacial heating mechanism, the nucleation and growth event on the GF could be controlled by programming the temperature profiles of the WIJH system. **Rapid switch of ramping and cooling could trigger and stop the crystallization, and the pulse temperature could be used to control the nucleation and growth with the synergy of the concentration effect.** Below is the detailed discussion.

1) Rapid ramping and cooling could control the crystallization process

Fig. R13 Temperature evolution of the WIJH system for the synthesis of HKUST-1/GF in 0.95 s. The red and gray regions represent the rapid ramping and cooling processes, respectively.

As the WIJH system presents high heating and cooling rates (e.g., a heating rate of up to 300 $^{\circ}\text{C s}^{-1}$ and a cooling rate of around 200 $^{\circ}\text{C s}^{-1}$ for the synthesis of HKUST-1/GF in 0.95 s (Fig. R13)), the crystallization could be triggered and stopped timely by the ramping and cooling, respectively. As shown in Fig. R14, the particle size increased from ~ 140 to ~ 854 nm, as the synthesis time increased from 0.25 to 0.95 s. This indicates that the rapid cooling could timely stop the crystallization by cutting down the electrified procedures with specific duration times. Moreover, the rapid switch of the ramping and cooling process (pulse cycle) could control the nucleation and growth. As shown in Fig. R15-a, as the temperature ramping-cooling cycled with periodic pulses, the nucleation events repeated, while the growth was suppressed due to the relatively low

temperature throughout. As a result, the normalized particle density increased from 1 particle per μm^2 for two pulses to (63.6 ± 10.9) particles per μm^2 GF for six pulses, while the particle size remained around 450 nm.

2) Higher-intensity pulse with high temperature promotes nucleation, while milder pulse benefits growth

Besides, as shown in Fig. R15-b, a high-intensity pulse with a high temperature in a short time (5.25 A, 300 °C, 0.25 s) would lead to burst nucleation, producing smaller particles around 491 nm. While under a low-intensity pulse with a low temperature for a long duration (1.85 A, 184 °C, 5 s), a few particles were produced in the nucleation portion, leaving sufficient reactive species unreacted. Afterward, the growth proceeded and became preferable over time. As a result, larger particles around 1281 nm were synthesized.

The above details and discussion have been added to the revised manuscript and supplementary information.

Fig. R14 SEM images of HKUST-1/GF obtained by WIJH with a) 0.25 s, b) 0.4 s, c) 0.55 s, d) 0.7 s, e) 0.85 s, f) 0.95 s. Scale bar: 2 μm .

Fig. R15 a) Pulse-cycle-based modulation towards particle density and b) pulse-intensity-based modulation towards particle size. From top to bottom are the electrified procedure (the current pattern), the temperature profile of the WIJH system, the statistical results of a) the particle density (normalized to the value of the sample prepared within 2 pulses) and b) the particle size, and the typical SEM images of the HKUST-1/GF (the border color of the image was used to mark the corresponding sample in the statistical result, scale bar: 1 μm for all).

Responses to Reviewer #3:

In this work, Zhang et al. reported a wet-interfacial Joule heating approach for synthesis of nanomaterials on conductive carbon films like graphene films. A very thin liquid layer is coated on the graphene film and the Joule heating rapid bring it to a high temperature to trigger the reaction. They synthesized various nanomaterials such as MOF, COF, metal nanoparticles, oxide nanoparticles, etc. And they demonstrated the application of the MOF films for CO₂ uptake and the application of the Joule heating method for temperature swing adsorption.

We sincerely thank the reviewer for the professional and careful comments. These insightful suggestions are very constructive to improve our work. We have tried our best to revise our manuscript accordingly, and the point-by-point responses are provided below.

Main concerns:

1. One main concern of this method is the scalability. The reaction happens at the interface between the graphene film and the solution; thus, the solution must be a very thin layer. This seriously limit its production rate and scalability compared to the bulk solution synthesis method. As demonstrated by the authors, a typical trial only uses 2 μL solution. Could the authors demonstrate some strategies to scale up the process? Maybe a continuous process?

R: Thank you. As the procedure is simple, and the devices are accessible, **the WIJH approach could be scaled up readily for the continuous and mass fabrication of nanomaterials with the aid of a roll-to-roll Joule-heating setup.** The practicability of the continuous fabrication of a fusing film and dispersed nanoparticles have been demonstrated, giving the production rates around 14.7 $\text{cm}^2 \text{s}^{-1}$ and 25 $\text{cm}^2 \text{s}^{-1}$. **The estimated production rate for the fabrication of nanomaterials-coated films (around 265 m h^{-1}) is superior to that reported by one of the latest advances in rapid production of MOFs ($\approx 4 \text{ m h}^{-1}$ in a batch experiment in lab scale, *Adv. Sci.* 7, 2002190, (2020)).** Note that 2 μL is a typical condition for synthesizing 0.2 cm^2 HKUST-1/GF. It is enough to achieve 100% coverage of HKUST-1 on the substrate to complete the preparation, and less requirement towards the precursors is one of the advantages of the WIJH synthesis. Besides, we have also developed **a layer-by-layer fashion to further control and increase the mass loading of the MOFs on the substrate** (from ~ 1.13 to $\sim 4.67 \text{ g m}^{-2}$), thereby improving the mass efficiency of the fabrication for potential industrial production. Below are the details:

The continuous fabrication system mainly consists of two pairs of parallel graphite electrodes (interval distance of 7 cm), two controllable micromotors to roll the electrodes with the speed of 10 rpm, and a d.c. power source (ITECH, IT65220) connecting with the electrodes via two electric brushes (Fig. R16-a). A continuous GF strip with a width of 2 cm was clamped and passed through the rotating electrodes. The precursor solution of HKUST-1 (85 mM $\text{Cu}(\text{NO}_3)_2$ and 55 mM H_3BTC) was spread on the GF to form a thin-layer liquid film (around 10 $\mu\text{L cm}^{-2}$, on the area between the electrodes). Afterward, an instantaneous current flow (24 A, 6 V) was applied to the electrodes to conduct the WIJH synthesis. Finally, the film was collected, followed by washing and drying.

For one batch of fabrication of HKUST-1/GF, both fusing HKUST-1 film (Fig. R16-b) and dispersed nanoparticles (Fig. R16-c) could be obtained by altering the electrified procedures, giving the production rates around 14.7 $\text{cm}^2 \text{s}^{-1}$ and 25 $\text{cm}^2 \text{s}^{-1}$, respectively. The estimated production rate for the fabrication of nanomaterials-coated films (around 265 m h^{-1}) is superior to that reported by one of the latest advances in rapid production of MOFs in industrial-level efficiency ($\approx 4 \text{ m h}^{-1}$ in a batch experiment in a lab scale, *Adv. Sci.* 7, 2002190, (2020)). Moreover, the fabrication could be

further scaled up by simultaneously increasing the area of the GF, the supplying power, and the volume of the precursors for one batch.

Furthermore, we have developed a layer-by-layer fashion to control and increase the mass loading and efficiency of the fabrication. The HKUST-1-coated GF was cycled into the area between two electrodes by the micromotors. A series of repeating procedures of adding the precursor solution on the same region of the GF and the WIJH synthesis were conducted successively until the fabrication was completed. MOF loading can be scaled up and easily adjusted through a repeating continuous layer-by-layer fashion, producing intergrown multi-layer HKUST-1 coatings on the GF (Fig. R16-d). From the first to five cycles, the mass rises from (1.13 ± 0.33) to (4.67 ± 0.38) g m⁻² (Fig. R16-e).

The above experimental details for continuous fabrication, results, and the corresponding discussion have been added in the revised manuscript and the supplementary information.

Fig. R16 a) Photographs of the setup for the roll-to-roll continuous fabrication for HKUST-1/GF. SEM images of a) the obtained fusing film and b) dispersed nanoparticles. c) SEM images of the multi-layer intergrown film obtained by the repeating layer-by-layer fashion. Inset is the cross-sectional SEM image. d) Areal loading of HKUST-1 on the GF in the layer-by-layer continuous fabrication.

2. The synthesized materials are loaded onto the graphene films substrates. Is there any appropriate method to separate the synthesized materials and the heating substrates?

R: Thank you. Using HKUST-1/GF as the model, **we have successfully peeled off HKUST-1 film from the GF substrate using adhesive tape** (Fig. R17). The tape was first pressed onto the coated HKUST-1 layer and removed from the GF surface in one quick swoop. This is a common method for mechanical exfoliation of materials (*Nat. Rev. Mater.* **6**, 605, (2021); *Adv. Mater.* **34**, 2202666, (2022)). The above result has been added in the revision.

Fig. R17 a) Schematic of the separation of the product of HKUST-1 film from the GF substrate by a tape, and b) photographs of the separated GF (left) and HKUST-1 film on the tape (right).

3. In Supplementary Table 2, the authors compared the energy cost, time consumption, etc., for a batch between the reported method here with literature routes. However, I recommend to normalize to the mass per batch since different literature routes obtained different amounts of materials. It's unfair to compare this method for 2 μL synthesis with other methods with, e.g., 2 mL synthesis.

R: Thank you. Following the reviewer's suggestion, we have collected information on the product mass in the literature, which was estimated according to the mass loading of MOFs and the mass of the substrate. Generally, the mass is variable via altering the dosage of the precursors, or the area of the substrate per batch (*Angew. Chem. Int. Ed.* **128**, 3480 (2016); *Adv. Sci.* **7**, 2002190, (2020)). The masses for one batch were 3.97-5.95 mg (*ACS Appl. Mater. Interfaces* **12**, 18437 (2020)), 155-930 mg (*Adv. Sci.* **7**, 2002190, (2020)), 1.2-3.9 mg (*Angew. Chem. Int. Ed.* **128**, 3480 (2016)), giving the production efficiencies of 0.046-0.069 $\mu\text{g s}^{-1}$, 86-520 $\mu\text{g s}^{-1}$, and 2-6.5 $\mu\text{g s}^{-1}$, respectively.

In our WIJH approach, the WIJH synthesis could be scaled up by a roll-to-roll continuous fabrication to give the mass per batch of 1.58-6.54 mg. The production efficiencies of 1376-1660 $\mu\text{g s}^{-1}$ were up to 10^4 higher than other methods. As for the energy cost based on the mass, WIJH also exhibited a low energy cost of around 1.07×10^{-4} kWh per mg MOFs) compared to other works (approximately 3.9×10^{-3} to 7.25 kWh per mg MOFs). Moreover, the mass of the product could be further increased by scaling up the substrate, increasing the concentration of the precursors, or increasing the volume of the precursors with the increase of the power for a batch. As mentioned above, the less requirement of the precursor solution is one of the advantages of WIJH synthesis. A microliter of the precursor solution is enough to achieve 100% coverage of nanomaterials on the substrate in our WIJH approach (e.g., 2 μL for 0.2 cm^2 GF). The large-volume bulk solution has been widely adopted in conventional methods. However, most of them are wasted for fabricating nanomaterials on the substrate.

Besides, **we also provided the comparison based on the area of the substrate** (Fig. 1h). The area information was facile to be collected in the experimental parts in the literature. More importantly, it could reflect the production efficiency objectively and comprehensively for fabricating nanomaterials on the substrate. The growth rate of $\sim 1.97 \mu\text{m s}^{-1}$ is up to 5 orders of magnitude faster than other typical heating-based methods (Table R2), while the energy cost of 9.55×10^{-6} kWh cm^{-2} is down to six orders of magnitude.

Table R2 Comparison of WIJH strategy used in this work for the synthesis of MOF films with other synthesis methods reported in the literature

Strategies	MOFs	Heating conditions		Synthesis efficiency for a batch			Refs.
		$T(^{\circ}\text{C})$	$t(\text{s})$	Particle size (nm)	Energy cost (kWh)	Production mass (mg)	
WIJH	HKUST-1	25-285,	0.95 0.25	~850 ~490	2.38×10^{-6} 1.91×10^{-6}	1.58-6.54	This work
Solvothermal synthesis	HKUST-1	110	43200	~20000	14.4-42	/	1
Solvothermal synthesis	HKUST-1	120	72000	/	2.63-3.63	/	2
Solvothermal synthesis	UiO-66-NH ₂	85	86400	50-100	28.8-84	3.97-5.95	3
Solvothermal synthesis	UiO-66	120	86400	150-500	2.86-4.23	/	4
Solvothermal synthesis	HKUST-1	120	201600	~2350	7.45-10.92	/	5
Microwave	ZIF-8	150	720	~800	1-1.8		
Microwave	MOF-5	120	30	~6500	4.16×10^{-3}	/	6
Microwave	UiO-66	120	1800	~300	0.0175-0.1	/	7
Microwave and evaporation	HKUST-1	90	1860	~170	0.6-1.75	/	8
Oven heating	HKUST-1	120	1800	~800	0.6-1.75	155-930	9
Spray-assisted method	HKUST-1	130	60	~1000	1.4×10^{-4}	/	10
Spray coating	ZIF-67	150	7200	44-5100	4-6	/	11
Vapor-assisted conversion	UiO-66	100	10800	~525	3.6-10.5	/	12
Thermal deposition	ZIF-8	200	900	~1000	0.3-0.88	/	13
Electrical induction heating	HKUST-1	141.3	28800	~2450	0.168	/	14
Hot-pressing	ZIF-8	200	600	~100	0.05-0.17	1.2-3.9	15

References

- [1] Y Sun, Y. et al. Oriented nano-microstructure-assisted controllable fabrication of metal-organic framework membranes on nickel foam. *Adv. Mater.* **28**, 2374-2381 (2016).
- [2] Lemaire, P. C. et al. Copper benzenetricarboxylate metal-organic framework nucleation mechanisms on metal oxide powders and thin films formed by atomic layer deposition. *ACS Appl.*

Mater. Interfaces **8**, 9514-9522 (2016).

[3] Yao, A., Jiao, X., Chen, D. & Li, C. Bio-inspired polydopamine-mediated Zr-MOF fabrics for solar photothermal-driven instantaneous detoxification of chemical warfare agent simulants. *ACS Appl. Mater. Interfaces* **12**, 18437-18445 (2020).

[4] Ghalei, B. et al. Rational tuning of zirconium metal-organic framework membranes for hydrogen purification. *Angew. Chem. Int. Ed.* **58**, 19034-19040 (2019).

[5] Liu, C. et al. General deposition of metal-organic frameworks on highly adaptive organic-inorganic hybrid electrospun fibrous substrates. *ACS Appl. Mater. Interfaces* **8**, 2552-2561 (2016).

[6] Yoo, Y. & Jeong, H. K. Rapid fabrication of metal organic framework thin films using microwave-induced thermal deposition. *Chem. Commun.* 2441-2443 (2008).

[7] Appelhans, L. N. et al. Facile microwave synthesis of zirconium metal-organic framework thin films on gold and silicon and application to sensor functionalization. *Micropor. Mesopor. Mat.* **323**, 111133 (2021).

[8] Ameloot, R. et al. Direct patterning of oriented metal-organic framework crystals via control over crystallization kinetics in clear precursor solutions. *Adv. Mater.* **22**, 2685-2688 (2010).

[9] Gao, G. K. et al. Rapid production of metal-organic frameworks based separators in industrial-level efficiency. *Adv. Sci.* **7**, 2002190 (2020).

[10] Kubo, M., Sugahara, T. & Shimada, M. Facile fabrication of HKUST-1 thin films and free-standing MWCNT/HKUST-1 film using a spray-assisted method. *Micropor. Mesopor. Mat.* **312**, 110771 (2021).

[11] Chen, Z. et al. Large-area crystalline zeolitic imidazolate framework thin films. *Angew. Chem. Int. Ed.* **60**, 14124-14130 (2021).

[12] Virmani, E. et al. On-surface synthesis of highly oriented thin metal-organic framework films through vapor-assisted conversion. *J. Am. Chem. Soc.* **140**, 4812-4819 (2018).

[13] Maina, J. W. et al. The growth of high density network of MOF nano-crystals across macroporous metal substrates – Solvothermal synthesis versus rapid thermal deposition. *Appl. Surf. Sci.* **427**, 401-408 (2018).

[14] Tao, Y., Huang, G., Li, Q., Wu, Q. & Li, H. Localized electrical induction heating for highly efficient synthesis and regeneration of metal-organic frameworks. *ACS Appl. Mater. Interfaces* **12**, 4097-4104 (2020).

[15] Chen, Y. et al. A Solvent-free hot-pressing method for preparing metal-organic-framework coatings. *Angew. Chem. Int. Ed.* **55**, 3419-3423 (2016).

4. The particle size of MOF synthesized by this method is a few hundreds of nanometer. What's the size controllability of this method? Could it be used to synthesize MOF nanoparticles (<100 nm) or large crystals?

R: Thank you. **The size controllability of the WIJH synthesis could be demonstrated as a wider variable size range from ~54 nm to ~1.65 μm, and a relatively uniform size distribution for each sample.** Stemming from the unique confined interfacial heating mechanism of the synergy between temperature effect and concentration effect, we could control the nucleation and growth of the nanomaterials on the GF, thereby customizing the product size. We have successfully obtained **small HKUST-1 nanoparticles of 54 ± 11 nm and large HKUST-1 microparticles of 1.65 ± 0.2 μm, through simultaneously adjusting the pulse intensity (i.e., the temperature) and the initial concentrations of the precursors.** Below is the detailed discussion:

Generally, the particle size is highly correlated to the crystallization process, which is affected by the reaction temperature, time, and concentration (*Chem. Rev.* **112**, 933, (2012); *Angew. Chem. Int. Ed.* **60**, 26390, (2021); *Nat. Chem.* **5**, 203, (2013)). According to the LaMél model, a rapid and intense nucleation event leads to small particles by depleting the precursors, while larger particles are obtained when growth is favorable. In the WIJH approach, a high-intensity pulse with high temperature enabled intense nucleation of MOFs, while a low-intensity pulse allowed a favorable growth event. Therefore, we realized the control of particle size via programming electrified procedures, obtaining particles with sizes ranging from 1281 ± 267 nm to 491 ± 100 nm by increasing the current intensities from 1.85 A to 5.25 A (Fig. 3b). On the other hand, as the initial concentration of the precursor of copper salt increases from 0.85 mM to 170 mM (with the same 3:2 molar ratio of $\text{Cu}(\text{NO}_3)_2$ and H_3BTC), the particle size increases from 207 ± 52 nm to 1208 ± 271 nm (Fig. 3c). Combining the above routes, we have successfully synthesized small HKUST-1 nanoparticles with the size of 54 ± 11 nm under a high-intensity pulse and low precursor concentrations (5.25 A, 0.85 mM $\text{Cu}(\text{NO}_3)_2$ and 0.56 mM H_3BTC , Fig. R18-a). Large microparticles with the size of 1.65 ± 0.2 μm were successfully synthesized under a low-intensity pulse and high precursor concentrations (1.85 A, 170 mM $\text{Cu}(\text{NO}_3)_2$ and 110 mM H_3BTC , Fig. R18-b).

Besides, benefitting from the ultrafast and successive nucleation and growth in the WIJH process, the particle was relatively uniform with a narrow size distribution. This contrasts sharply with the broad size distribution obtained by conventional solvothermal syntheses.

Fig. R18 SEM images of a) HKUST-1 nanoparticles of 54 ± 11 nm, and b) HKUST- microparticles of 1.65 ± 0.2 μm .

5. The authors demonstrate the CO_2 uptake and liberation cycle by the Joule heating process (Figure 5). What is the mechanism of the Joule heating-induced liberation?

R: Thank you. **The CO_2 uptake and liberation by HKUST-1/GF are based on a temperature swing adsorption process, in which the adsorption occurs at low temperature, while the desorption occurs at high temperature with a supply of electrified procedure to the HKUST-1/GF.** HKUST-1 could strongly adsorb CO_2 with the electrostatic interactions under low temperature. It is an exothermic physical process. The kinetic energies of the adsorbed CO_2 and HKUST-1 are higher within the rapid thermal conduction from the GF to HKUST-1 under Joule heating. This causes a corresponding increase in their interaction at the interface, thereby reducing the effective area of HKUST-1 available for adsorption (*ACS Appl. Mater. Interfaces* **10**, 34802 (2018)). Therefore, as shown in Fig. R19, the uptake would decrease in the CO_2 adsorption isotherm, and the amount of adsorbed CO_2 decreases as the temperature increases (i.e., a higher electrified input toward HKUST-1/GF). After cutting off the current, the temperature of the GF and HKUST-1 dropped sharply, and thus, HKUST-1 could adsorb CO_2 again, completing a desorption-adsorption cycle.

The mechanism for the Joule-heating-based liberation of the adsorbed CO₂ has been added in the revised manuscript.

Fig. R19 CO₂ adsorption isotherm of HKUST-1/GF under the programmed IJH processes.

6. What is the energy cost of the Joule heating induced CO₂ desorption and how does it compare to temperature swing enabled by other thermal processes, as well as other pressure swing process?

R: Thank you. In an interfacial Joule heating (IJH) process based on HKUST-1/GF, **the energy for the desorption of 1 cm³ CO₂ was estimated to be 3.72×10^{-4} kWh** (the full desorption of 9.2 cm³ g⁻¹ CO₂ with power of 5.95 w for 200 s, green region in Fig. R19). **It is lower or comparable with other previously-reported thermal processes** (*Sep. Purif. Technol.* **309**, 123053 (2023)); *ChemSusChem* **13**, 2089 (2020)).

Generally, **temperature swing adsorption is considered a low-cost and energy-saving route** for the desorption of CO₂, **when compared with other swing processes**, e.g., pressure swing adsorption (*Acc. Chem. Res.* **50**, 778 (2017); *Chem. Eng. J.* **383**, 123075 (2020)). However, the low thermal conductivity of the adsorbents, e.g., most microporous MOFs, would hinder this process and strongly increase energy consumption (*Adv. Mater.* **28**, 1839, (2016)). In our interfacial Joule heating strategy, the Joule heat around GF was directly transferred to the CO₂-adsorbed MOFs. The direct and compact contact between GF and HKUST-1 could sharply shorten the heat transfer distances, improving desorption efficiency. As for the comparison with other thermal processes, e.g., microwave, and magneto-thermal processes, **Joule heating is known as a simple, highly-efficient, and energy-saving heating strategy.** GF possesses ultrahigh electrothermal conversion efficiency (>99%), due to the low heat capacity and appropriate electrical conductivity. It sharply decreases energy consumption. Moreover, the IJH strategy presents high controllability towards the adsorption and desorption, which is hard to realize in conventional methods.

The above discussion has been added to the revised manuscript.

7. Will the precursors also be evaporated during the synthesis process since the synthesis temperature is so high (>300°C)? And the organics maybe be destroyed under this high temperature.

R: Thank you. **The precursors of organics and salts would not be evaporated or decomposed.** In our WIJH approach, the heating temperature of the WIJH system was set to be less than the thermal decomposition temperature of the product to avoid thermal destruction. **The ultrahigh temperature (e.g., 300 °C) only occurs at the graphene film interface, while precursors exist in the solution part where the temperature is lower than their respective decomposition temperatures.** Using the synthesis of HKUST-1 as the model, we have supplemented the thermogravimetric analysis (TGA) to investigate the thermal decomposition behaviors of both precursors and product, and collected XRD patterns to confirm there are not any decomposed

products of the precursors in the final products.

Fig. R20-a shows the TGA curves of the precursors of H₃BTC and Cu(NO₃)₂·2.5H₂O and the product of HKUST-1/GF. H₃BTC and HKUST-1 possess high decomposition temperatures around 340 °C and 300 °C, respectively. Cu(NO₃)₂·2.5H₂O exhibits two distinct stages of mass loss ranging from 132 to 176.2 °C and 207.6 to 290.2 °C, which are ascribed to the dehydration and the thermal decomposition to form copper oxide (*J. Therm. Anal. Calorim.* **147**, 5599 (2022); *J. Therm. Anal.* **45**, 1381 (1995)). As shown in Fig. R20-b, in the case of the growth of HKUST-1 on the GF, the temperature profile of the WIJH system ranges from ~25 to ~285 °C, which comprises the temperature of the GF (~25 to 288 °C) and the solution layer (~25 to ~153 °C). Within the ultrafast consumption of the precursors for crystallization, the precursors dissolved in the solution part throughout. As its highest temperature (153 °C) is lower than the initial thermal decomposition temperatures of Cu(NO₃)₂ (207.6 °C) and H₃BTC (300 °C), precursors would not be decomposed. Besides, the coordination between precursors spontaneously occurs after mixing. Their strong interactions would increase the thermal stabilities of the precursors, and thus, reduce their volatilization, evaporation, and even decomposition (*Nat. Commun.* **14**, 2294 (2023)).

Furthermore, no characteristic peaks indexed to the precursors' decomposition products were found in the XRD pattern of HKUST-1/GF (Fig. 1g), which further indicated that precursors were not decomposed.

The above results have been discussed in the revision.

Fig. R20 TGA curves in air and the temperature profiles during a typical WIJH synthesis for HKUST-1 (the electrified procedure of 3A for 0.95 s).

Minor points:

1. Supplementary Fig. 5 is not discussed in the main text.

R: Thank you, and sorry for the negligence. Supplementary Fig. 5 was involved in the supplementary discussion 3, and we have marked the supplementary Fig. 5 in the revision.

2. Line 217, the authors mentioned that the particles could fuse together to form a film consisting of monolayer MOF particles. There is no evidence that the film is “monolayer”.

R: Thank you. As shown in Fig. R21, the cross-sectional SEM image of the HKUST-1/GF indicates that the HKUST-1 film was composed of one layer of fusing HKUST-1 particles. To avoid the ambiguity for monolayer, we have updated the expression as “the film consisting of one layer of fusing MOF particles” in the revised manuscript.

Fig. R21 Cross-sectional SEM image of the HKUST-1/GF.

3. PDF reference card should be provided for all the XRD patterns.

R: Thank you. All PDF reference cards for the XRD patterns have been provided in the revised manuscript and supplementary information.

REVIEWERS' COMMENTS

Reviewer #1 (Remarks to the Author):

All the previous concerns have been carefully addressed by the authors and the quality of the manuscript has been greatly improved. Thus, this paper has my recommendation for acceptance.

Reviewer #2 (Remarks to the Author):

The authors have addressed my comments properly. It is ok for me to accept the revised manuscript for publishing.

Reviewer #3 (Remarks to the Author):

The authors have resolved most of my concerns. I think it is acceptable after the authors addressing some remaining smaller issues.

1. The authors clarified the novelty of this work compared to existing works, that is, the confined interfacial heating between liquid and the carbon heater. This makes it differentiated from other solid state Joule heating syntheses. While the explanation in the response letter is comprehensive, I suggest the authors discuss more on the Introduction part of the manuscript to give credit to previous works in the field (e.g., 10.1038/s41467-020-20084-5, 10.1038/s41467-022-32622-4). BTW, Line 47, the authors wrote "in solid-solid systems, ... , value-added C2 products⁷". Note that ref. 7 (10.1038/s41586-022-04568-6) is a solid-gas system.
2. The authors demonstrated the roll-to-roll process for scaling up, which is promising for mass production. Could the author provide some more characterization of the upscaled sample? Is the material uniform across the substrate?
3. Figure R18a, the particle size is too small on this scale. I suggest zoom-in on the SEM image to provide better resolution.
4. PDF card should be provided in the figure caption for all the XRD patterns in Fig. 1g, Fig. 4f-h.
5. Figure 1c inset, scale bar required for the sample size.

RESPONSE TO REVIEWERS' COMMENTS

Reviewer #1:

All the previous concerns have been carefully addressed by the authors and the quality of the manuscript has been greatly improved. Thus, this paper has my recommendation for acceptance.

R: We are sincerely grateful for your recommendation of our manuscript for publication in *Nature Communications*.

Reviewer #2:

The authors have addressed my comments properly. It is ok for me to accept the revised manuscript for publishing.

R: We appreciate the reviewer for recommending our manuscript for publication in *Nature Communications*.

Reviewer #3:

The authors have resolved most of my concerns. I think it is acceptable after the authors addressing some remaining smaller issues.

R: We are sincerely grateful for the positive and valuable comments on the manuscript, and we further revised the manuscript point-by-point.

1. The authors clarified the novelty of this work compared to existing works, that is, the confined interfacial heating between liquid and the carbon heater. This makes it differentiated from other solid state Joule heating syntheses. While the explanation in the response letter is comprehensive, I suggest the authors discuss more on the Introduction part of the manuscript to give credit to previous works in the field (e.g., 10.1038/s41467-020-20084-5, 10.1038/s41467-022-32622-4). BTW, Line 47, the authors wrote “in solid-solid systems, ... , value-added C2 products⁷”. Note that ref. 7 (10.1038/s41586-022-04568-6) is a solid-gas system.

R: Thank you. Following the reviewer’s suggestion, we have provided more detailed discussions in the introduction part (marked in red in the revised manuscript). More previous significant works in Joule-heating-based synthesis have been cited (*Science* **368**, 521-526 (2020); *Science* **376**, eabn3103 (2022); *Nat. Commun.* **13**, 5027 (2022); *Nat. Commun.* **14**, 2294 (2023)). The novelty of WIJH approach has been highlighted, and the differences between previous solid-state systems and our solid-liquid system have been emphasized, particularly in thermal conduction and crystallization mechanism. Besides, we have revised the description in line 47 as “it enables a series of generally unachievable processes in solid-solid and/or solid-gas systems, such as the synthesis of high-entropy nanoparticles, gram-scale graphene in milliseconds, and value-added C2 products with high selectivity”.

2. The authors demonstrated the roll-to-roll process for scaling up, which is promising for mass production. Could the author provide some more characterization of the upscaled sample? Is the material uniform across the substrate?

R: Thank you. We have further supplemented other characterizations for the upscaled sample, including XRD and SEM. XRD pattern (Figure R1-b) confirmed the production of pure crystalline

HKUST-1 by the roll-to-roll Joule heating method. To evaluate the uniformity of the HKUST-1 film across the substrate, we have collected SEM images from different sites along the length direction of the upscaled sample (the distance of each collecting site was about 2 cm). As shown in Figure R1-a, all the samples exhibited similar morphologies of flat fusing films, indicating a relatively uniform distribution of the nanomaterials in a long range.

The above results and discussion have been added in the supplementary information in red.

Figure R1 Characterizations of the products obtained by roll-to-roll continuous fabrication.

a) SEM images of the samples from different sites along the length direction of the upscaled fusing HKUST-1/GF (the distance of each collecting site was about 2 cm), scale bar: 1 μm . b) XRD patterns of simulated HKUST-1 (gray) and the fusing HKUST-1/GF (red) within different 2θ ranges.

3. Figure R18a, the particle size is too small on this scale. I suggest zoom-in on the SEM image to provide better resolution.

R: Thank you. The figure has been replaced by magnified SEM images of the sample to display the particle size (Figure R2, i.e., Supplementary Fig. 26).

Figure R2 SEM images of HKUST-1 nanoparticles on the GF obtained by WIJH.

4. PDF card should be provided in the figure caption for all the XRD patterns in Fig. 1g, Fig. 4f-h.

R: Thank you. PDF cards for all XRD patterns in the manuscript and supplementary information have been added in the corresponding figure caption parts.

5. Figure 1c inset, scale bar required for the sample size.

R: Thank you. Figure 1c (Figure R3) has been updated as follows, in which the scale bar of the inset has been added.

Figure

Figure R3 Temperature evolution of the GF with customized electrified procedures. Inset is a typical thermal image during a Joule heating process (3 A for 1 s, scale bar: 0.1 cm).